# THE UBIQUITY OF 2-HOMOGENEITY, HOW ITS IMPLICIT BIAS SELECTS FEATURES, AND OTHER STORIES

## ABSTRACT

This work studies the optimization and generalization consequences of a seemingly innocuous design choice in many modern architectures: they end with a composition of affine parameters belonging to a normalization layer and a linear layer, resulting in a fundamentally 2-homogeneous architecture. The first set of results are abstract, showing how any architecture satisfying this type of 2-homogeneity and a few regularity conditions on the gradients of the inner layers obtain *large margins* and low test error. As technical byproducts, this part of the story provides an *implicitly biased gradient flow* guarantee and also a nondecreasing margin lemma for *inhomogeneous* networks. The second set of results instantiate this framework for shallow normalized ReLU networks, establishing large margin and low test error via feature selection purely from random initialization and standard gradient flow. As a corollary, the paper obtains good test error for $k$-bit parity problems, in particular passing below sample complexity lower bounds from linearized analyses such as the *Neural Tangent Kernel*.

## 1 INTRODUCTION

The surprising beneficial properties of first-order optimization methods remain one of the main great mysteries of deep learning: while a vast body of literature arose trying to understand how optimization seems to find parameters which seemingly overfit but still generalize well (Neyshabur et al., 2014; Zhang et al., 2016), practice has continued apace. For instance, recent work identifies even more sophisticated variants of the phenomenon in modern settings such as LLMs, where optimization automatically yields low-dimensional structures with many beneficial properties (Li et al., 2018; Aghajanyan et al., 2020; Hu et al., 2021).

For classification tasks where near-perfect test accuracy is possible, such as image classification (Krizhevsky, 2009), a classical mathematical analysis comes via the concept of *prediction margins*, where the differences between carefully normalized output logits directly yields an upper bound on generalization error (Bartlett, 1996; Schapire et al., 1997). While this classical theory is rooted in linear predictors and dates back to the original Perceptron analysis (Novikoff, 1962), it nevertheless has predictive power within modern deep learning; for example, it gives an explanation for test accuracy continuing to drop after perfect training accuracy is reached, a phenomenon which resurfaced as part of the *grokking* story (Power et al., 2022).

Closer to the present work, a sensitive margin-based analysis can illuminate feature learning and its consequences on generalization error. One concrete approach of this flavor was nicely laid out by Bach (2017), who gave two function classes implicitly searched during deep network optimization: the simpler class is $\mathcal{F}_2$, which roughly speaking corresponds to a lack of feature learning, such as exhibited by the *Neural Tangent Kernel (NTK)* (Jacot et al., 2018), and a more complex class $\mathcal{F}_1$, which allows for feature learning, but seems generally computationally intractable to search over. Concordantly, while it is reasonably easy to show that gradient descent can compete with the best function in $\mathcal{F}_2$ (Ji & Telgarsky, 2020b; Chen et al., 2019), competing with the best functions in $\mathcal{F}_1$ has only been shown under a combination of modified algorithms and stringent assumptions which are either hard-to-establish or even potentially false (Wei et al., 2018; Chizat & Bach, 2020). Even so, probing how far gradient descent can venture into $\mathcal{F}_1 \setminus \mathcal{F}_2$ remains an effective way to study feature learning and the generalization benefits of first-order methods.

The goal of the present work is to show how the normalization structure present in modern architectures can aid in feature learning and good test error, first presenting an abstract mathematical framework in Section 3, and then specializing this framework to shallow normalized ReLU networks in Section 4. A summary of the high-level contributions is as follows.

1. The first contribution is an elementary observation: Section 3.1 outlines how the aforementioned product structure of modern normalized architectures aids in searching over $\mathcal{F}_1$. Section 1.1 will also provide a teaser, but the phenomenon is simple enough: optimizing $\mathcal{F}_1$ corresponds to identifying good but rare features, thus has a sparsity flavor, and meanwhile the product structure causes gradient descent to instead follow an $\ell_1$ geometry, which is classically understood as sparsity-preferring.

2. The first substantial technical contribution comes in the remainder of Section 3 (also teased in Section 1.1), showing two consequences of this product structure for abstract architectures: **(a)** how the *implicit bias of gradient descent*, combined with the product structure and a few conditions on the gradients of inner layers, gives rise to optimization over $\mathcal{F}_1$, which previously was only establish with modified algorithms and extensive assumptions (Wei et al., 2018; Chizat & Bach, 2020); and **(b)** how the search over $\mathcal{F}_1$ does not degrade over time, extending the seminal work of Lyu & Li (2019) to *inhomogeneous* networks such as transformers.

3. The second contribution, appearing in full in Section 4 but again teased in Section 1.1, is a concrete instantiation of the previous framework for shallow normalized ReLU networks, in particular using an idealization of BatchNorm (Ioffe & Szegedy, 2015), extremely large width, but no other assumptions or modifications. As shallow ReLU networks have been the subject of great study, this analysis gives explicit test error bounds which are shown to beat test error lower bounds for well-studied problems such as the *k-parity problems*; by contrast, prior work needs a combination of algorithmic assumptions and distributional assumptions (Barak et al., 2022; Glasgow, 2023).

The remainder of this introduction teases the results; Sections 3 and 4 contained the details and proof sketches; the appendices contain full details.

## 1.1 MAIN RESULT TEASER

This subsection provides a slightly more formal teaser of the main results, detailed in Sections 3 and 4. Throughout this work, the prediction problem will be binary classification with binary labels $y \in \{\pm 1\}$, gradient descent will be idealized as the gradient flow, and the loss will be the logistic loss $\ell(z) := \ln(1 + \exp(-z))$ (with corresponding average $\mathcal{L}(w) := \frac{1}{n} \sum_i \ell(y_i h(x_i; w)))$. While the ReLU and various other architectural choices are not differentiable, this work uses the standard choice of replacing the gradient by the minimum norm element of the Clarke differential (Lyu & Li, 2019; Ji & Telgarsky, 2020a), as detailed further in Section 3.1.

**Normalization and $2$-homogeneity.** The following trivial observation gives the basis for the present work. Standard normalization layers (e.g., BatchNorm, LayerNorm, and RMSNorm (Ioffe & Szegedy, 2015; Ba et al., 2016; Zhang & Sennrich, 2019)) all come with a multiplicative parameter which is enabled by default in modern deep learning toolkits (Paszke et al., 2019). Moreover, modern architectures, such as the transformers used in LLMs (Touvron et al., 2023), conclude with such a normalization layer followed by a linear layer, with no intermediate activation. This effect is idealized in this work as a *product* of two linear layers, treating the whole network mapping as

$$x \mapsto \sum_{j=1}^{m} a_j b_j F_j(x; V) = \langle a \odot b, F(x; V) \rangle,$$

where outer parameters $((a_j, b_j))_{j=1}^m$ correspond to the final linear layer and adjacent multiplicative normalization parameters, and the mappings $(F_j)_{j=1}^m$ with shared parameters $V \in \mathbb{R}^p$ correspond to the behavior of the inner layers, such as an MLP and attention stack in a transformer. The second form introduces some convenient notation, where now $(a, b) \in \mathbb{R}^m \times \mathbb{R}^m$ and $a \odot b$ is their coordinate-wise product, and $F : \mathbb{R}^d \times \mathbb{R}^p \to \mathbb{R}^m$ collects the individual internal functions $(F_j)_{j=1}^m$ into a big multivariate mapping, which after all better captures the behavior of a modern architecture with inner parameters $V \in \mathbb{R}^p$. *(add olmo architecture)*

As will be detailed in Section 3.1, gradient flow on these parameters is *equivalent* to a *mirror flow* (an infinitesimal variant of mirror descent, just as gradient flow is infinitesimal gradient descent), in particular a mirror flow on a new set of parameters which firstly obeys an $\ell_1$ geometry, and secondly is convex in the new single outer parameter after this reparameterization. This reparameterization therefore solves two problems: the $\ell_1$ geometry leads to optimization over $\mathcal{F}_1$, and the convexity gives some hope of tractability. By contrast, the problem before reparameterization is a second degree polynomial in $(a, b)$ and not convex.

**Abstract mathematical framework.** The 2-homogeneity alone is hopeful but does not yield any concrete results; these come via additional structural assumptions and beneficial *implicit biases* of gradient flow and the reparameterized mirror flow.

As mentioned before, the approach here to feature learning and good test error is through the lens of margins, summarized as follows. Let $h(x; w)$ denote a binary classifier with input example $x$ and parameters $w$; for example, the main 2-homogeneous setup here has the form $h(x; (a, b, V)) = \langle a \odot b, F(x; V) \rangle$. The unnormalized margin is simply $yh(x)$, and it is clear that predictions are correct when this quantity is positive, but how to meaningfully interpret its magnitude and a corresponding notion of confidence? In particular, one must adjust for the fact that scaling up $(a, b)$ would amplify this unnormalized margin. Mathematically, classical margin theory establishes good generalization but only after a careful choice of *normalization* by some function of the model parameters: for classical coordinate-descent based boosting methods such as AdaBoost, $\ell_1$, namely $yh(x; w)/\|w\|_1$, was the appropriate normalization (Schapire et al., 1997); for gradient descent, $\|w\|_2$ in the denominator makes sense (Soudry et al., 2017; Ji & Telgarsky, 2018); for ReLU networks of depth $L$, the correct normalization seems to be $\|w\|_2^L$ (Lyu & Li, 2019).

A ReLU network with $L$ layers is an example of an $L$-homogeneous architecture; the full definition will come in Section 3, but a key point is that standard modern choices such as transformers violate homogeneity in a vast assortment of ways. The present work provides a way out and a corresponding baseline: essentially all the mathematical tools of homogeneity go through if one restricts attention to a *subset* of the parameters which obey homogeneity, even while the study is of gradient descent on the *original full set of parameters*.

Writing intuitively until the formal development in Section 3, consider a network which is $L$-homogeneous in a subset $\hat{w}$ of the full parameters $w$; for instance, the above framework is 2-homogeneous in parameters $(a, b)$. The corresponding margin definition simply ignores the other parameters:

$$\text{margin}(\hat{w}) := \frac{\min_i y_i h(x_i; w)}{\|\hat{w}\|^L}.$$

Of course, this definition is only meaningful if the numerator can not grow arbitrarily with the norm of the ignored parameters $w \setminus \hat{w}$; that will also be a property of this mathematical treatment and critically affect the generalization properties of 2-layer ReLU networks discussed shortly and in Section 4.

The result will require a few more explanations: of the initial good features, slow-growing derivatives, and the smooth margin; these will come after the statement, and in detail in Section 3.

**Theorem 1.1** (Informal simplification of Lemma 3.2 and theorems 3.1 and 3.6)**.** *1. (**Early good margins**.) Suppose there exist* initial good features with margin $\gamma$ *and the gradient flow exhibits* slow-growing derivatives*; then there exists a time $t$ such that*

$$\frac{y_i h(x_i; w(t))}{\|a(t)\|_2^2 + \|b(t)\|_2^2} \geq \frac{\gamma}{16}.$$

*2. (**Asymptotic nondecreasing margins**.) Suppose $h$ is $L$-homogeneous in a subset $\hat{w}$ of the full parameter set $w$, and $w$ is updated with gradient flow on $\mathcal{L}$. If there exists a time $t$ with $\mathcal{L}(w(t)) < \ell(0)/n$, then $\mathcal{L}(w(s)) \to 0$ and the smooth margin $\ell^{-1}(n\mathcal{L}(w(s)))/\|\hat{w}(s)\|^L$ is nondecreasing.*

*In particular, in the 2-homogeneous framework above, $\frac{\ell^{-1}(n\mathcal{L}(w(s)))}{\|a(s)\|^2 + \|b(s)\|^2}$ is nondecreasing.*

Summarizing the results, part 1 of theorem 1.1. covers the initial phase of GF, namely GF achieves a large margin, and part 2 covers the late regime of GF (large margin is maintained thereafter). Of

note, part 1 only applies to architectures with a specific outer layer structure (eq. (2)) whereas part 2 handles more general partially homogeneous architectures.

Before discussing the concrete instantiation of Theorem 1.1 to shallow normalized ReLU networks, the various confusing pieces can be sketched with appropriate pointers Section 3. Firstly, the worrisome *slow-growing derivatives* property is easily to satisfy with architecture choices such as BatchNorm, though a more extensive study (beyond Section 4) is needed and pushed to future work. The *initial good features* assumption may sound as though this analysis performs no feature learning, and indeed it is better described as *feature selection* or a reweighting of good initial features; the mathematical analysis here establishes a sanity check that no matter what, enough feature *stay still* so that one can still choose good outer weights, though it also allows for widely-changing features. Lastly, the *smooth margin* $\ell^{-1}(n\mathcal{L}(w))$ may seem quite complicated, but is a standard object in the analysis of margins (Lyu & Li, 2019; Schapire et al., 1997; Telgarsky, 2013), and becomes the hard margin in this setting as $t \to \infty$. While these various conditions may seem stringent, they are easily satisfied as below for shallow normalized ReLU networks, and moreover are enough to exit $\mathcal{F}_2$ (and thus the Neural Tangent Kernel) and optimize over $\mathcal{F}_1$.

**Concrete instantiation: normalized shallow ReLU networks.** Theorem 1.1 was ultimately hopelessly abstract; this next teaser gives a completely concrete instantiation which removes all the nebulous assumptions.

The architecture in this case will be an idealization of BatchNorm (Ioffe & Szegedy, 2015): it suffices to carve $V \in \mathbb{R}^p$ into $m$ vectors $(v_j)_{j=1}^m$ with $v_j \in \mathbb{R}^d$ and define $F_j(x; V) := \sigma(\langle v_j/\|v_j\|, x\rangle)$, giving an overall predictor

$$x \mapsto \langle a \odot b, F(x; V)\rangle = \sum_j a_j b_j \sigma\left(\left\langle \frac{v_j}{\|v_j\|}, x\right\rangle\right). \tag{1}$$

The notion of optimal $\mathcal{F}_1$ margin in this setting is well-studied (Bach, 2017; Chizat & Bach, 2020), and while it will appear in detail in Section 4, the point here is that it is strong enough to yield test error guarantees, indeed ones beating $\mathcal{F}_2$ in many interesting cases.

**Theorem 1.2** (See Section 4 for setting details.)**.** *Let $h$ correspond to the shallow normalized architecture in eq.* (1)*, Suppose the* data distribution *has margin $\gamma$ (cf. Section 4). Then for all $t \geq t_0$ and $m \geq m_0$ (where $t_0$ depends on $\gamma$ and $m_0$ depends on $\gamma$ and $\ln(t)$), with probability at least $1 - \delta$ over the draw of random initial parameters $w(0) := (a(0), b(0), V(0))$ and data $((x_i, y_i))_{i=1}^n$, both the empirical margins and test error are good:*

$$\min_i \frac{y_i h(x_i; w(t))}{\|a(t)\|_2^2 + \|b(t)\|_2^2} \geq \frac{\gamma}{32}, \qquad \Pr\left[yh(x; w(t))\right] \leq \widetilde{\mathcal{O}}\left(\frac{1}{n\gamma^2}\right).$$

*In particular, for the* support-only *$k$-parity problems (cf. Section 4), the test error is $\lesssim \frac{dk^2}{n}$ whereas for the NTK (one perspective on $\mathcal{F}_2$) it is $\gtrsim \frac{d^k}{n}$.*

To conclude this teaser subsection, a few explanations are in order. Firstly, the abstract assumptions of Theorem 1.1 have been translated into the lower bounds $(m_0, t_0)$; unfortunately, these lower bounds are prohibitively large and should be taken as *essentially* infinite (they are double exponential). On the other hand, the closest result in the literature has both quantities as *truly* infinite, and makes additional hard-to-verify assumptions which are circumvented in this work (Chizat & Bach, 2020).

Secondly, to demonstrate a concrete separation between $\mathcal{F}_1$ and $\mathcal{F}_2$, the statement concludes with test error rates for $k$-parity problems, which have received tremendous study (Barak et al., 2022; Glasgow, 2023). The strongest result for an unmodified network is due to Glasgow (2023), and obtains a strong result in the sense of allowing a narrow network and small number of training iterations, but on the other hand the distributional assumption is much more strict, requiring only the 2-parity problem and not allowing the more difficult "support-only" variant of Section 4.

## 2 PRELIMINARIES

**Architecture** Given $a, b \in \mathbb{R}^m$ and $V \in \mathbb{R}^p$ and $F : \mathbb{R}^d \times \mathbb{R}^p \to \mathbb{R}^m$, Section 3.1 considers general models $h$ of the form

$$h(x; w = (a, b, V)) = \langle a \odot b, F(x; V)\rangle, \tag{2}$$

where $F$ satisfies the following assumption.

**Assumption 2.1.** The feature map $F : \mathbb{R}^d \times \mathbb{R}^p \to \mathbb{R}^m$ is bounded,

$$\sup_x \sup_V \left\| F(x; V) \right\|_\infty < \infty.$$

The bounded assumption is rather mild as all features maps that end with a normalization layer such as Layer Normalization or Root Mean Square Normalization satisfy this property. For initialization, let weights be initially distributed with $a_j, b_j \sim \mathrm{Unif}(\pm 1/\sqrt{m})$, and $V$ be drawn from an arbitary distribution. Section 4 considers shallow ReLU networks and hence the feature map $F$ take the form

$F_j(x; V) := \sigma \left( \left\langle v_j / \|v_j\|, x \right\rangle \right)$ which certainly satisfies assumption 2.1.

**Gradient Flow.**    As mentioned, this work uses the gradient flow on all parameters, namely $w(t)$ is the solution to the differential equation

$$\frac{\mathrm{d}}{\mathrm{d}t} w(t) = \dot{w}(t) = -\nabla_w \mathcal{L}(w(t)).$$

Since the networks in general will not be differentiable, formally the flow is a solution to the *Clarke differential inclusion* $\dot{w}(t) \in -\partial_w \mathcal{L}(w(t))$, where $\partial_w$ denotes the Clarke differential Lyu & Li (2019). This differential has many intricacies, but this work tacitly assumes all networks are locally Lipschitz and o-minimal definable, which suffices to guarantee chain rules and remove the main difficulties (Lyu & Li, 2019; Ji & Telgarsky, 2020a); due to this and the well-established nature of the Clarke differential in deep learning theory, this work will simply write $\nabla$ in place of $\partial$ to aid readers unfamiliar with the Clarke differential, who are directed for more background to (Lyu & Li, 2019; Ji & Telgarsky, 2020a).

**Vector Operations.**    Given any scalar valued univariate function $f$ and vector $v$, the value $f(v)$ is defined to be the vector where $f$ is applied elementwise. In a similar fashion, for any scalar $r \in \mathbb{R}$ and vector $v$, inequalities $v \geq r$ is taken to be understood as each entry of $v$ being at least $r$.

## 3    ABSTRACT, ARCHITECTURE-INDEPENDENT ANALYSIS

This section first formally restates Theorem 1.1 while providing proof sketches along the way.

### 3.1    RESTATEMENT OF THEOREM 1.1

**Theorem 3.1** (Formal restatement of part 1 of Theorem 1.1)**.** *Let $t$ be given, define*

$$C_1 := \sup_{s<t} \sup_{i,r,j} \left| \left\langle \nabla_V F_j(x_i; V(s)), \nabla_V F_j(x_r; V(s)) \right\rangle \right|,$$

$$C_2 := \sup_{s<t} \sup_{i,r,j \neq k} \left| \left\langle \nabla_V F_j(x_i; V(s)), \nabla_V F_k(x_r; V(s)) \right\rangle \right|,$$

*and let $\bar{p}$ and $\gamma$ be given with $\|\bar{p}\|_1 = 1$ and $\min_i \left\langle \bar{p}, y_i F(x_i; V(0)) \right\rangle \geq \gamma$, and lastly define $u(0) := a(0) \odot b(0)$ and $R := \mathrm{KL}(\bar{p}, u(0))$. If these various quantities satisfy*

$$\|\bar{p}\|_\infty \leq \frac{\gamma^3}{4 C_1 \left[ \ln(tn)(8R+2) + 32 \right]^2}, \qquad C_2 \leq \frac{\gamma}{4 \left[ \ln(tn)(8R+2) + 32 \right]^2},$$

*then if $t \geq \left( \frac{20R}{\gamma} \right)^4$,*

$$\min_i \frac{y_i h(x_i; w)}{\|a(t)\|^2 + \|b(t)\|^2} \geq \frac{\gamma}{4(1+R)}.$$

*If additionally $t \geq e + \exp(\exp(4nR/\gamma))$, then*

$$\min_i \frac{y_i h(x_i; w)}{\|a(t)\|^2 + \|b(t)\|^2} \geq \frac{\gamma}{32}.$$

Hence, given any concrete instantiation of the model $h$, Theorem 3.1 reduces the problem of obtaining large margins to checking growth of the gradient of the feature maps and existence of initial good features which is embodied by $(\bar{p}, \gamma)$.

Looking back to Theorem 1.1, here $(\bar{p}, \gamma)$ capture the "initial good features" property, and $C_1$ and $C_2$ capture the "slow-growing derivatives" property. As will be seen in Section 4, $(\bar{p}, \gamma)$ are well-studied for shallow ReLU networks, and meanwhile $C_1$ can be made arbitrarily small with large width, and $C_2$ is simply 0. With more sophisticated architectures, $C_2$ becomes interesting; arguably one of the main byproducts of the NTK theory is that a quantity similar to $C_2$ goes to 0 as width goes to infinity (Jacot et al., 2018), which would be interesting and valuable to establish in the present setting.

The quantity $R$ can be quite bad; e.g., in Section 4, in the worst case it seems to scale as $d \ln(1/\gamma)$, whereas the best possible results truly require just a constant such as $\gamma/16$. On the other hand, pushing $R$ doubly-exponentially into $t$ is also unsavory. This time lower bound is mostly likely a technical artifact which will be discussed in the material to come. Before delving into the proof sketch of theorem 3.1, some preliminaries, concerning the reparameterized flow mentioned in Section 1 and general implicitly-biased mirror descent guarantees, must be established.

## 3.2 Reparameterized Flow and Implicitly-biased Mirror flow guarantees

As it stands, the current gradient flow is not easy to analyze with respect to the outer layers $(a, b)$, since even if there is some beneficial 2-homogeneous structure, the objective is not jointly convex in this pair. To make the analysis more tractable, consider the following reparameterization: given a model $h$ with corresponding weights $w = (a, b, V)$, let the reparameterized weights be $(u, V)$ where

$$u := |a \odot b|.$$

The following lemma shows that the reparameterized weights are updated via an appropriate *mirror flow* along with other useful identities.

**Lemma 3.2.** *Let $h(x; w = (a, b, V))$ correspond to the architecture described by eq. (2) and $h$ is trained via gradient flow. At initialization, suppose $|a_j(0)| = |b_j(0)|$ for all $j \in [m]$. Abbreviate $\beta_j := \mathrm{sgn}(a_j(0)b_j(0))$. Then, for all time $t$,*

$$u_j(t) = a_j(t)^2 = b_j(t)^2 = \beta_j a_j(t) b_j(t),$$

*and additionally,*

$$\frac{\mathrm{d}}{\mathrm{d}s} \ln u(s) = -\nabla_u 2\mathcal{L}\left(\left\langle u \odot \beta, F(\cdot; V)\right\rangle\right) = -2\nabla_u \mathcal{L}(w(s)). \tag{3}$$

This is the reparameterized flow, and many remarks are in order. Firstly, this is called *mirror flow* because the time derivative is not on $u(s)$ directly, but on $\ln u(s)$. Here, $\ln(\cdot)$ is a *mirror map*, and induces a different geometry. This geometry is well-studied and quite special, it induces a sparsity structure (Shalev-Shwartz et al., 2011). The lemma establishes a correspondence between the update on $(a, b, V)$ and on $(u, Q)$ and show that $(a, b)$ inherits the beneficial sparsity-inducing structure enforced directly on $u$ via the mirror map, which leads to optimization over $\mathcal{F}_1$ rather than $\mathcal{F}_2$. The uniqueness of 2 homogeneity becomes apparent when analyzing a more general $L$ homogeneous model of the form $h(x; w = (a^1, \ldots a^L, V)) = \left\langle a^1 \odot \cdots \odot a^L, F(x; V)\right\rangle$. Defining the reparamterized weights as $u_j(t) := \prod_{\ell=1}^{L} a_j^\ell(t)$, one can quickly determine that eq. (3) fails for $L \neq 2$. Hence, only 2-homogeneous models explicitly optimize over $\mathcal{F}_1$.

Note that versions of this connection have been made before when $a \odot b$ is replaced by $a \odot a$ (cf. (Chizat & Bach, 2020, Section 4), and (Woodworth et al., 2020)), the difference here will be an explicit connection with $a \odot b$, the remark that this arises from normalized structure, and all the upcoming optimization consequences of this setup; in particular, prior work only obtained $\mathcal{F}_1$ results when freezing the inner layers and their corresponding parameters $V$, whereas this work allows them to be updated with standard gradient flow (Chizat & Bach, 2020, Section 4).

The key observation is that the function $u \mapsto 2\mathcal{L}\left(\left\langle u \odot \beta, F(\cdot; Q)\right\rangle\right)$ is *convex in $u$*. This convexification allows an immediate invocation of mirror descent tools; these tools are strengthened here (in Lemma 3.3) to identify an implicit bias, which powers all the results of this section.

Thanks to the exact equivalence given in Lemma 3.2, the remainder of this section will be stated *entirely* in the re-parameterized flow, which shortly will be shown to fit very nicely within the framework of online convex optimization, with an added upcoming twist of implicit bias. Specifically, as a final pieces of notation, define

$$z_i(s) := y_i \beta \odot F(x_i; V(s)), \qquad \mathcal{L}_s(u) := \frac{1}{n} \sum_i \ell(\langle u, z_i(s) \rangle); \tag{4}$$

by this choice, the prediction problem begins to look like a standard online convex optimization problem with *time-varying objective* $\mathcal{L}_s$, but convex in the relevant parameter $u(s)$; indeed, it appears to be a simple linear prediction problem with examples $(z_i(s))_{i=1}^n$ varying over time. The remainder of this subsection details implicitly-biased mirror flow bound for a generic mirror flow setup. Let $\phi$ denote a *Legendre map* — basically the corresponding mirror map $\nabla \phi$ and its inverse $\nabla \phi^*$ are guaranteed to always be well-defined, for details see for instance (Lattimore & Szepesvári, 2020) — and let $D_\phi$ denote the corresponding *Bregman divergence*

$$D_\phi(u, v) = \phi(u) - \left[ \phi(v) + \langle \nabla \phi(v), u - v \rangle \right].$$

Given a time-varying potential $(f_s)_{s \geq 0}$, the general mirror flow is defined as the solution to the differential equation

$$\frac{\mathrm{d}}{\mathrm{d}s} \nabla \phi(u(s)) = -\nabla f_s(u(s)).$$

The general implicitly-biased guarantee and its specialization to the setup sparsity (entropy) here are as follows.

**Lemma 3.3.** *1. Suppose the general mirror flow setup and that each $f_s$ is convex. For any $t \geq 0$ and any $u(0)$ and any $\bar{u}$,*

$$\min_{s < t} \left[ f_s(u_s) - f_s(\bar{u}) \right] \leq \frac{D_\phi(\bar{u}, u(0)) - D_\phi(\bar{u}, u(t))}{t}.$$

*2. Suppose $\phi(u) = \langle u, \ln(u) - 1 \rangle = \sum_j \left( u_j \ln u_j - u_j \right)$. Then for any $u, v \in \mathrm{dom}(\phi) = \mathbb{R}_{>0}^m$,*

$$\nabla \phi(u) = \ln u,$$

$$D_\phi(v, u) = \left\langle v, \ln \frac{v}{u} \right\rangle + \langle 1, u - v \rangle =: \mathrm{KL}(v, u),$$

*and moreover for any $t \geq 0$, any $\bar{u}, u(0) \in \mathrm{dom}(\phi) = \mathbb{R}_{>0}^m$, then*

$$\frac{\|u(t)\|_1}{8} \mathbf{1} \left[ \|u(t)\|_1 \geq 2\|\bar{u}\|_1 \right] + \int_0^t f_s(u(s)) \, \mathrm{d}s \leq \mathrm{KL}(\bar{u}, u(0)) + \int_0^t f_s(\bar{u}) \, \mathrm{d}s.$$

To recover the reparameterized flow, by lemma 3.2 and recalling the notation described by eq. (4), it suffices to set the mirror map and time varying potential as

$$\nabla \phi(u) = \ln(u), \quad \text{and} \quad f_s(u) = -2\mathcal{L}_s(u).$$

With the mirror flow tools at hand, a proof sketch of Theorem 3.1 can be given.

### 3.3 Proof sketch of theorem 3.1

Using the implicitly-biased mirror flow guarantees, one can establish simultaneous control over the norms of the predictors $u(t)$ as well as their loss $\mathcal{L}_s(u_s)$. Such guarantees are sufficient to guarantee that the predictor $u(t)$ has positive but potentially small margin. In order to obtain a large margin guarantee, more fine grained analysis is needed by splitting the proof into two regimes: the early phase where the norms of the predictors $u(t)$ are small and the asymptotic phase where the norms of the predictors $u(t)$ are large (i.e. $\mathcal{O}(\frac{\ln(tn)}{\gamma})$). For the remainder of the section, fix the reference solution $\bar{u}$ as $\bar{u} := \frac{\ln(tn)}{\gamma} \bar{p}$ and denote

$$G(s) := -\frac{1}{n} \sum_{i=1}^n \ell'(\langle u(s), z_i(s) \rangle),$$

which will appear in many places in the analysis. An interesting and critical property of the logistic loss is the *self-bounding* property $G(s) \leq \mathcal{L}_s(u(s))$.

Under the theorem constraints on time $t$ and width $m$, one can argue that the $\ell_1$ unit reference solution $\bar{p}$ remains a good linear predictor over the feature maps even across time.

$$\inf_{s \in [0,t]} \min_{i \in [n]} \langle \bar{p}, z_i(s) \rangle \geq \frac{\gamma}{2}. \tag{5}$$

The proof of the preceding fact is rather technical but at a high level involves arguing that simultaneously $\int_0^t G(s) \, \mathrm{d}s$ and $\|V(t) - V(s)\|$ is small. Instantiating Lemma 3.3 for the logistic and our entropy-based mirror flow by setting

$$\nabla \phi(u) = \ln(u), \quad \text{and} \quad f_s(u) = -2\mathcal{L}_s(u),$$

and applying the *self-bounding* property $G(s) \leq \mathcal{L}_s(u(s))$ results in the following lemma.

**Lemma 3.4.** *Let $t$ be given, and suppose there exists $\bar{p} \geq 0$ and $\gamma > 0$ with $\|\bar{p}\|_1 = 1$ and $\inf_{s \in [0,t]} \min_i \langle \bar{p}, z_i(s) \rangle \geq \gamma$. Define $\bar{u} := \frac{\ln(tn)}{\gamma} \bar{p}$, and $R := D_\phi(\bar{u}, u(0)) / \|\bar{u}\|_1$. Then*

$$\frac{\|u(t)\|_1}{8} \mathbf{1} \left[ \|u(t)\|_1 \geq 2\|\bar{u}\|_1 \right] + 2 \int_0^t G_s \, \mathrm{d}s \leq D_\phi(\bar{u}, u(0)) + 2 \int_0^t \mathcal{L}_s(\bar{u}) \, \mathrm{d}s$$

*and moreover $\|u(t)\|_1 \leq \|u(0)\| \exp \left( 2 \int_0^t G_s \, \mathrm{d}s \right)$, and*

$$\sup_{s \leq t} \|u(s)\|_1 \leq (8R + 2)\|\bar{u}\|_1 + 32, \qquad 2 \int_0^s G_r \, \mathrm{d}r \leq 2 \int_0^s \mathcal{L}_r(u(r)) \, \mathrm{d}r \leq R\|\bar{u}\|_1 + 2.$$

*The following further guarantees hold under additional assumptions.*

1. *If $t \geq t_1 := \left( \frac{20R}{\gamma} \right)^4$, then*

$$\min_i \frac{\langle z_i(t), u(t) \rangle}{\|u(t)\|_1} \geq \frac{\gamma}{2(1 + R)}.$$

2. *If $t \geq t_1$ and $\|u_t\|_1 \leq C\|\bar{u}\|_1$ with $C \leq 4 + R$, then*

$$\min_i \frac{\langle z_i(t), u(t) \rangle}{\|u(t)\|_1} \geq \frac{\gamma}{2C}.$$

To establish the first margin guarantee in theorem 3.1, by eq. (5), part a of lemma 3.4 can be applied to get the desired margin bound. Now for the large margin guarantee, as mentioned in the opening, consider the following two cases.

1. **Early Phase.** Suppose $\|u(t)\| \leq 8\|\bar{u}\|$. By case assumption, part b of Lemma 3.4 can be applied with $C = 8$; hence, MF guarantees tailored to logistic loss and entropy-based mirror flow immediately gives the desired hard margin result in the early phase,

$$\min_i \frac{\langle z_i(t), u(t) \rangle}{\|u(t)\|_1} = 2 \min_i \frac{y_i h(x_i; w)}{\|a(t)\|^2 + \|b(t)\|^2} \geq \frac{\gamma}{16}.$$

   The factor 2 stems from the fact $2\|u(t)\|_1 = \|a(t)\|^2 + \|b(t)\|^2$.

2. **Asymptotic Phase.** On the other hand, suppose $\|u(t)\| > 8\|\bar{u}\|$. To tackle this case, the following lemma is needed.

   **Lemma 3.5.** *Suppose $t \geq t_1$ as defined in Lemma 3.4. assume $\|u(0)\|_1 = 1$ and let $\bar{p}$ be given as before.*

$$\frac{\psi_t(u_t)}{\|u_t\|_1} \geq \gamma - \frac{1 + 2\ln(n)}{\|u_t\|_1} - \frac{\mathrm{KL}(\bar{p}, u(0))}{\xi_t}, \tag{6}$$

   *where*

$$\xi_t = \int_0^t \ell'(\psi_s(u_s)) \, \mathrm{d}s \geq \frac{1}{n} \int_0^t G(s) \, \mathrm{d}s \geq \frac{1}{n} \ln \|u_t\|_1.$$

By case assumption, $\|u(t)\| > 8\|\bar{u}\| = 8\frac{\ln(tn)}{\gamma}$. By hypothesis, $t \geq e + \exp(\exp(4nR/\gamma))$. Applying the lower bound for $t$ and the lower bound $\xi_t \geq \frac{1}{n}\ln\|u_t\|_1$ to eq. (18) grants the desired margin guarantee. The lower bound $\xi_t \geq \frac{1}{n}\ln\|u_t\|_1$ indicates the dominant term in the margin maximization rate is $\frac{\mathrm{KL}(\bar{p},u(0))}{\xi_t}$. Indeed, it is this term that forces time $t$ to be doubly exponential. The proof of Lemma A.3 involves expanding the standard mirror flow potential function $D_\phi(\bar{u}, u(t)) - D_\phi(\bar{u}, u(0))$, substituting the gradient of the loss with the scaled gradient of the smoothed margin, applying convexity, and some algebra. The proof additionally makes use of nondecreasing margin theorem theorem 3.6 which shall described in detail in the following section.

## 3.4 NONDECREASING MARGINS FOR PARTIALLY-HOMOGENEOUS MODELS

This section provides a rigorous restatement of the second part of Theorem 1.1 and provides proof sketches. Before doing so, formally define *smooth margin* as

$$\psi(w) := \ell^{-1}(n\mathcal{L}(w)), \qquad \psi_s(u) := \ell^{-1}(n\mathcal{L}_s(u)).$$

This quantity indeed appears quite daunting and technical to start, so consider the simplifying case of $\ell(r) = \exp(-r)$, the exponential loss. In this case, $\psi_s(u) = -\ln\sum_i \exp(-\langle u, z_i(s)\rangle)$, and thus by usual properties of $\ln\sum\exp$ it is related to the hard margin $\min_i \langle u, z_i(s)\rangle$.

To restate the second part of Theorem 1.1, all that remains is homogeneity. A function is $L$-homogeneous if $f(cx) = c^L f(x)$ for any $c \geq 0$; thus our original parameterization is clearly 2-homogenous over $(a, b)$, a single ReLU is 1-homogeneous, and an $L$-layer ReLU network is $L$-homogenous over the full set of parameters. In this work, if some *coordinate-wise subset* of the parameters $\hat{w} \subseteq w$ induces homogeneity, meaning $f(c\hat{w}; w \setminus \hat{w}) = c^L f(\hat{w}; w \setminus \hat{w})$ for $c \geq 0$, then simply say $f$ is $L$-homogeneous with respect to parameters $\hat{w}$. Of course, gradient flow is still performed over all variables, even though the homogeneity definition ignores $w \setminus \hat{w}$. Even so, the seminal claim of nondecreasing margins from (Lyu & Li, 2019) goes through despite the potentially severe inhomogeneity of $w \setminus \hat{w}$.

**Theorem 3.6** (Restatement of part 2 of Theorem 1.1). *Suppose $h$ is $L$-homogeneous with respect to parameters $\hat{w} \subseteq w$, and that there exists a time $\tau$ with $\mathcal{L}(w(\tau)) < \ell(0)/n$. Then, for all $t \geq \tau$,*

$$\frac{\mathrm{d}}{\mathrm{d}t}\frac{\psi(w(t))}{\|\hat{w}(t)\|^L} \geq 0, \qquad \frac{\mathrm{d}}{\mathrm{d}t}\|\hat{w}(t)\| > 0.$$

*Abbreviate the potentially inhomogenous parameters as $\theta := w \setminus \hat{w}$. Additionally, if $\sup_\theta \sup_{\|\hat{w}\| \leq 1}\|\nabla_{\hat{w}} h(x; (\hat{w}, \theta))\| < \infty$, then*

$$\mathcal{L}(w(t)) \to 0, \quad \|\hat{w}(t)\| \to \infty.$$

*In particular, if $h(x; w) := \langle a \odot b, F(x; V)\rangle$ where $F$ is a bounded function, and also letting $(u, V)$ denote the corresponding reparameterized mirror flow, then similarly $\mathcal{L} \to 0$ and*

$$\frac{\mathrm{d}}{\mathrm{d}t}\frac{\psi(w(t))}{\|a(t)\|^2 + \|b(t)\|^2} = \frac{\mathrm{d}}{\mathrm{d}t}\frac{\psi(w(t))}{2\|u(t)\|_1} \geq 0, \qquad \frac{\mathrm{d}}{\mathrm{d}t}\|a(t)\| = \frac{\mathrm{d}}{\mathrm{d}t}\|b(t)\| > 0, \qquad \frac{\mathrm{d}}{\mathrm{d}t}\|u_s\|_1 > 0.$$

Several remarks are in order. Firstly, the theorem holds for *any* model $h$ that has a subset of homogeneous parameters and not just models of the form $h(x; w = (a, b, V)) = \langle a \odot b, F(x; V)\rangle$. Additionally, the nondecreasing *normalized* margin and norm guarantees do not require boundedness of partial derivatives with respect to the homogeneous parameter $\hat{w}$. Of course, the notion of margin only makes sense when $|h|$ does not grow arbitrarily large as the norm of the potential inhomogeneous parameters go to infinity. Compared to Lyu et al. (2021, lemma 5.2), theorem 3.6 applies to various classes of inhomogeneous functions including ResNet and transformer models.

Comparing proof techniques, Lyu et al. (2021) analyzes the logarithm of the smoothed margin by decomposing into its radial and tangential components. It is not clear how to use the radial and tangential decomposition argument when there exists inhomogeneous parameters. To avoid using this decomposition, the proof of theorem 3.6 deals directly with the smoothed margin. In detail, the proof starts by brute-force expanding $\frac{\mathrm{d}}{\mathrm{d}s}\frac{\psi(w(s))}{\|\hat{w}(s)\|^L}$. Various terms arise of the form

$$\langle \nabla\psi(w(s)), \dot{w}(s)\rangle = \|\nabla\psi(w(s))\| \cdot \|\dot{w}(s)\| \geq \|\nabla_{\hat{w}}\psi(w(s))\| \cdot \|\dot{\hat{w}}(s)\|,$$

where the equality followed from colinearity of $\nabla\psi(w(s))$ and $\nabla\mathcal{L}(w(s))$ and $\dot{w}(s)$ and the lower bound followed simply by dropping coordinates which necessarily shrinks the norm; applying this in a few places converts $w$ to $\hat{w}$ and allows the application of homogeneity. Miraculously, homogeneity with respect to the entire $w$ is never needed. Details are in the appendices.

## 4 SHALLOW ReLU NETWORKS

This section will provide sufficient detail to complete the setting of Theorem 1.2, sketch its proof, and provide additional context. As mentioned, this section will feature a *distributional* margin assumption. Namely, let $\mathcal{D}$ denote a distribution over $(x, y)$ pairs and $\mathrm{supp}(\mathcal{D})$ its support. Let $\mathcal{P}_d$ denote the set of signed measures on $\{v \in \mathbb{R}^d : \|v\| = 1\}$ with unit mass (for details on this standard setup, see (Bach, 2017; Chizat & Bach, 2020)); this setup can be taken as an infinite-dimensional abstraction of $\ell_1$ bounded outer weight vectors. Define the appropriate distributional $\mathcal{F}_1$ margin (matching (Chizat & Bach, 2020)) as

$$\gamma_{\mathcal{D}} := \sup_{\mu \in \mathcal{P}_d} \inf_{(x,y) \in \mathrm{supp}(\mathcal{D})} y \int \sigma(v^{\mathsf{T}} x) \, \mathrm{d}\mu(v).$$

which implies $\int \sigma(v^{\mathsf{T}} x) \, \mathrm{d}\mu(v) \geq \bar{\gamma}_{\mathcal{D}}/2$ for $\mathcal{D}$-almost-every $(x, y)$.

In order to apply the tools of Theorem 3.1 in the present finite-width setting of eq. (1), this definition must be *subsampled* to produce a good candidate $\bar{p}$ at initialization. In particular, let weights be initially distributed with $a_j, b_j \sim \mathrm{Unif}(\pm 1/\sqrt{m})$, and $v_j$ uniform on the surface of the sphere (this is a *mean-field initialization* as it gives $u(0) = (1/m, \ldots, 1/m)$). The details of this sampling are standard, and can be found in the appendices. All that remains is to check the various constants in Theorem 3.1; by calculus, $C_1 \leq 1$ and $C_2 = 0$. Thus, with the candidate $\bar{p}$ and constants $C_1, C_2$ in hand, the following theorem can be established.

**Theorem 4.1.** *Let $h$ correspond to the shallow normalized architecture in eq. (1). Suppose data $((x_i, y_i))_{i=1}^n$ is drawn from a given distribution $\mathcal{D}$ and the $\mathcal{F}_1$ margin of shallow networks on $\mathcal{D}$ is $\gamma > 0$. If $t \geq \left(\frac{20R}{\gamma}\right)$ and width $m \geq \max\left\{\frac{2^{10} \ln^2(tn)d^2}{\gamma^8}, 2^{10}\ln(2n/(\delta\gamma))^2 \left(\frac{8}{\gamma}\right)^{2d-2}\right\}$ with probability at least $1 - \delta$ over the draw of the random initial parameters $w(0) := (a(0), b(0), V(0))$ and data $((x_i, y_i))_{i=1}^n$, the empirical margin and test error satisfy,*

$$\min_i \frac{y_i h(x_i; w_t)}{\left\|(a(t), b(t))\right\|^2} \geq \frac{\gamma}{32d\ln(\frac{1}{\gamma})}, \quad \Pr\left[y h(x; w(t))\right] \leq \widetilde{\mathcal{O}}(\frac{d^2}{n\gamma^2}).$$

*Additionally, there exists a time $t_0 > 0$ and width $m_0 > 0$ such that for all $t \geq t_0$ and $m \geq m_0$, with probability at least $1 - \delta$ over the draw of the random initial parameters $w(0) := (a(0), b(0), V(0))$ and data $((x_i, y_i))_{i=1}^n$, the empirical margin and test error satisfy,*

$$\min_i \frac{y_i h(x_i; w_t)}{\left\|(a(t), b(t))\right\|^2} \geq \frac{\gamma}{32}, \quad \Pr\left[y h(x; w(t))\right] \leq \widetilde{\mathcal{O}}(\frac{1}{n\gamma^2}).$$

To make the theorem results more concrete consider the *support-only $k$-parity problem* is as follows.

**Example 4.1** (Support-only $k$-parity problem). *Consider any distribution whose input margin is supported on the corners of the normalized hypercube, meaning $x \in \{-1/\sqrt{d}, +1/\sqrt{d}\}^d$, and the label $y$ is given by the product of someone unknown set $S$ of $k = |S|$ bits: $y = d^{k/2} \prod_{j \in S} x_j$. This is a classical problem in machine learning theory which has re-awoken to study feature learning in deep networks (Wei et al., 2018; Barak et al., 2022; Glasgow, 2023). The $\mathcal{F}_1$ margin of this problem is known to be at least $\frac{1}{k\sqrt{d}}$ (Telgarsky, 2022), whereas the $\mathcal{F}_2$ margin is $\frac{1}{d^{k/2}}$, with corresponding sample complexity lower bounds (Barak et al., 2022); plugging these claims into the general setup of Theorem 1.2 gives the stated sample complexities. While it is true that some works manage a much stronger result than the one here, for instance Glasgow (2023) is able to show that with width and time just poly-logarithmic in $d$, gradient descent can learn the 2-bit parity problem, all mentioned prior works require the margin distribution on the corners of the normalized hypercube to be uniform (Wei et al., 2018; Barak et al., 2022; Glasgow, 2023), hence motivating the "support-only" name here. On the one hand, it makes the results here more general, but on the other, this potentially leads to the blow-up in width and time; indeed, it is an interesting question if it is possible to do better without restricting the marginal distribution.*

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

## A   Deferred proofs from Section 3

First comes an extremely technical lemma establishing basic properties of the losses; the final property is partly to blame for the explosively high bound on $t$. Let $\ell_{\log}(z) := \ln(1 + \exp(-z))$ and $\ell_{\exp} = \exp(-z)$ denote logistic and exponential loss respectively.

**Lemma A.1.** *Let $(z_i)_{i=1}^n$ with $z_i \in \mathbb{R}^d$ be given, and define $Z \in \mathbb{R}^{n \times d}$ with $i^{\text{th}}$ row $z_i^{\mathsf{T}}$. Denote $\mathcal{L}_{\exp}(u) := \frac{1}{n} \sum_{i=1}^n \ell_{\exp}(\langle u, z_i \rangle)$ and $\mathcal{L}_{\ell_{\log}}(u) := \frac{1}{n} \sum_{i=1}^n \ell_{\log}(\langle u, z_i \rangle)$.*

1. *If loss $\ell$ is nonincreasing and nonnegative, then*

$$\min_i \langle \bar{u}, z_i \rangle \geq \psi(Z\bar{u}).$$

2. *For $\ell \in \{\ell_{\log}, \ell_{\exp}\}$, if $\min_i \langle \bar{u}, z_i \rangle \geq 0$, then*

$$\min_i \langle \bar{u}, z_i \rangle \leq \psi(Z\bar{u}) + c_1 \ln n + c_2,$$

   *where $(c_1, c_2) = (2, 1)$ for $\ell_{\log}$ and $(c_1, c_2) = (1, 0)$ for $\ell_{\exp}$.*

3. *For $\ell \in \{\ell_{\log}, \ell_{\exp}\}$, defining $\gamma := \min_i \frac{\langle \bar{u}, z_i \rangle}{\|\bar{u}\|_1}$, then*

$$\mathcal{L}(\bar{u}) \leq \exp(-\gamma \|\bar{u}\|_1).$$

4. *Fix $u \in \mathbb{R}^m$. Suppose there exists constants $C_2, C_3, C_5 > 0$ such that $\|u\|_1 \leq C_2 \|\bar{u}\|_1$, and $\mathcal{L}_{\exp}(u) \leq C_5 \exp(-\gamma C_2 \|\bar{u}\|_1 / C_3)$ and $\|\bar{u}\|_1 \geq \frac{2C_3}{\gamma C_2} \ln(2nC_5)$. Then*

$$\min_i \frac{\langle u, z_i \rangle}{\|u\|_1} \geq \frac{\gamma}{2C_3}.$$

5. *For any $v \in \mathbb{R}^n$, defining $G(v) := \frac{1}{n} \sum_i |\ell'(\langle v, z_i \rangle)|$,*

$$G(v) \leq -\ell'(\psi(v)) \leq \min\left\{1, en^2 G(v)\right\}.$$

*Proof.*   1. Since $\ell$ is nonincreasing, so is $\ell^{-1}$, and by nonnegativity of $\ell$,

$$\min_i \langle \bar{u}, z_i \rangle = \min_i \ell^{-1}\left[\ell(\langle \bar{u}, z_i \rangle)\right] = \ell^{-1}\left[\max_i \ell(\langle \bar{u}, z_i \rangle)\right] \geq \ell^{-1}\left[\sum_i \ell(\langle \bar{u}, z_i \rangle)\right] = \psi(Z\bar{u}).$$

2. Starting from the same inequalities as before, but using the nonincreasing property in the other direction,

$$\min_i \langle \bar{u}, z_i \rangle = \ell^{-1}\left[\max_i \ell(\langle \bar{u}, z_i \rangle)\right] \leq \ell^{-1}\left[\frac{1}{n} \sum_i \ell(\langle \bar{u}, z_i \rangle)\right].$$

   To proceed further, for the exponential loss, $\ell^{-1}(z) = -\ln(z)$, so

$$\ell^{-1}\left[\frac{1}{n} \sum_i \ell(\langle \bar{u}, z_i \rangle)\right] = -\ln \frac{1}{n} \sum_i \ell(\langle \bar{u}, z_i \rangle) = \psi(Z\bar{u}) + \ln n.$$

   On the other hand, for the logistic loss, combining (Ji & Telgarsky, 2019, Lemma 5.4) and (Ji & Telgarsky, 2020a, Lemma C.5), letting $q_i$ denote the dual variables, since $\min_i \langle z_i, \bar{u} \rangle \geq 0$,

$$\min_i \langle \bar{u}, z_i \rangle \leq \frac{1}{\|q\|_1} \sum_j q_j \min_i \langle \bar{u}, z_i \rangle \leq \frac{1}{\|q\|_1} \sum_i q_i \langle \bar{u}, z_i \rangle \leq \frac{\psi(Z\bar{u}) + 2 \ln n + 1}{\|q\|_1}$$

$$\leq \psi(Z\bar{u}) + 2 \ln n + 1.$$

3. The two cases can be proved simultaneously, noting

$$\mathcal{L}_{\log}(\bar{u}) = \frac{1}{n} \sum_i \ln(1 + \exp(-\langle \bar{u}, z_i \rangle))$$

$$\leq \frac{1}{n} \sum_i \exp(-\langle \bar{u}, z_i \rangle)$$

$$= \mathcal{L}_{\exp}(\bar{u})$$

$$\leq \frac{1}{n} \sum_i \exp(-\min_j \langle \bar{u}, z_j \rangle)$$

$$\leq \exp(-\gamma \|\bar{u}\|_1).$$

4. Abbreviate $X_i := \langle u, z_i \rangle$ and let $X$ be the random variable drawn uniformly over values $\{X_i\}_{i=1}^n$. By Markov's inequality, and since norm $\|u\|_1 \leq C_2 \|\bar{u}\|_1$ and also that $\ell_{\exp}$ is nonincreasing,

$$\sum_i \mathbf{1}\left[\frac{\langle u, z_i \rangle}{\|u\|_1} \leq \frac{\gamma}{2C_3}\right] = \sum_i \mathbf{1}\left[\langle u, z_i \rangle \leq \frac{\gamma \|\bar{u}\|_1}{2C_3}\right]$$

$$\leq \sum_i \mathbf{1}\left[\langle u, z_i \rangle \leq \frac{\gamma C_2 \|\bar{u}\|_1}{2C_3}\right]$$

$$= \sum_i \mathbf{1}\left[\exp(-\langle u, z_i \rangle) \geq \exp\left(-\frac{\gamma C_2 \|\bar{u}\|_1}{2C_3}\right)\right]$$

$$= n \Pr\left(\exp(-X) \geq \exp\left(-\frac{\gamma C_2 \|\bar{u}\|_1}{2C_3}\right)\right)$$

$$\leq n \frac{\mathbb{E}\left[\exp(-X)\right]}{\exp\left(-\frac{\gamma C_2 \|\bar{u}\|_1}{2C_3}\right)}$$

$$= n \frac{\mathcal{L}_{\exp}(u)}{\exp\left(-\frac{\gamma C_2 \|\bar{u}\|_1}{2C_3}\right)}. \tag{7}$$

By hypothesis, $\mathcal{L}_{\exp}(u) \leq C_5 \exp(-\gamma C_2 \|\bar{u}\|_1 / C_3)$ and $\|\bar{u}\|_1 \geq \frac{2C_3}{\gamma C_2} \ln(2nC_5)$. Applying the preceding inequalities to eq. (7),

$$\sum_i \mathbf{1}\left[\frac{\langle u, z_i \rangle}{\|u\|_1} \leq \frac{\gamma}{2C_3}\right] \leq \frac{n\mathcal{L}_{\exp}(\bar{u})}{\exp(-\gamma C_2 \|\bar{u}\|_1 / (2C_3))}$$

$$= n\mathcal{L}_{\exp}(\bar{u}) \exp\left(\gamma C_2 \|\bar{u}\|_1 / C_3 - \gamma C_2 \|\bar{u}\|_1 / (2C_3)\right)$$

$$\leq \frac{1}{2C_5} \mathcal{L}_{\exp}(\bar{u}) \exp\left(-\gamma C_2 \|\bar{u}\|_1 / (2C_3)\right) < 1,$$

which implies $\sum_i \mathbf{1}\left[\frac{\langle u, z_i \rangle}{\|u\|_1} \leq \frac{\gamma}{2C_2}\right] = 0$.

5. Furthermore, since $\ell(r) = \ln(1 + \exp(-r))$, then $\ell^{-1}(s) = -\ln(\exp(s) - 1)$, and $\ell'(r) = -(1 + \exp(r))^{-1}$, abbreviating $p_i := \langle v, x_i \rangle$,

$$\ell^{-1}\left(\sum_i \ell_i(p_i)\right) = -\ln\left(\exp\left[\sum_i \ln(1 + \exp(-p_i))\right] - 1\right)$$

$$= -\ln\left(\prod_i (1 + \exp(-p_i)) - 1\right)$$

$$= -\ln\left(\sum_{\substack{S \subseteq [n] \\ |S| \geq 1}} \prod_{i \in S} \exp(-p_i)\right)$$

$$=: -\ln M,$$

$$\ell'(\psi(v)) = \frac{-1}{1 + \exp(-\ln(M))}$$

$$= \frac{-M}{1 + M}$$

$$\in (-1, 0).$$

Take index $j \in \arg\min_{i \in [n]} p_i$. Then

$$-\ell'(\psi(v)) = \frac{1}{1 + \frac{1}{M}}$$

$$\geq \frac{1}{1 + \frac{1}{\exp(-p_j)}}$$

$$= -\ell'(p_j)$$

$$\geq -\frac{1}{n} \sum_i \ell'(p_i).$$

For the upper bound, consider two cases. If $p_j \geq \ln(n)$, then

$$-\ell'(\psi(v)) = \sum_i \frac{\exp(-p_i) \sum_{\substack{S \subseteq [n] \setminus \{i\} \\ |S| \geq 0}} \prod_{j \in S} \exp(-p_j)}{1 + M}$$

$$\leq \sum_i \frac{\exp(-p_i) \sum_{\substack{S \subseteq [n] \setminus \{i\} \\ |S| \geq 0}} n^{-|S|}}{1 + M}$$

$$\leq \sum_i \frac{\exp(-p_i) \sum_{j \geq 0} \binom{n}{j} n^{-j}}{1 + M}$$

$$\leq \sum_i \frac{\exp(-p_i) \sum_{j \geq 0} \frac{1}{j!}}{1 + \exp(-p_i)}$$

$$= e \sum_i -\ell'(p_i).$$

On the other hand, if there exists $p_j < \ln(n)$, then

$$-\ell'(\psi(v)) \leq 1$$

$$\leq \frac{2n}{1 + \exp(\ln(n))}$$

$$\leq \frac{2n}{1 + \exp(p_j)}$$

$$\leq -2n \sum_i \ell'(p_i).$$

$\square$

Next, some bounds on the unbounded Kullback-Leibler divergence KL used throughout; recall the definition

$$\mathrm{KL}(p, q) := \left\langle p, \ln \frac{p}{q} \right\rangle + \langle 1, q - p \rangle,$$

which agrees with the standard KL over the simplex, but handles the entire orthant.

**Lemma A.2.** *Fix Legendre potential* $\phi(p) := \langle p, \ln(p) - 1 \rangle$.

1. *Note* $\nabla\phi(v) = \ln(v)$, *and* $D_\phi(a,b) = \mathrm{KL}(a,b) = \langle a, \ln(a/b)\rangle - \langle 1, a-b\rangle$, *and* $\min_{v\in\Delta_m} D(v,b) = -\ln\|b\|_1$, *which is attained uniquely by the choice* $b/\|b\|_1$ *(whereby in the notation of the two-stage mirror descent, $w_{i+1} = v_{i+1}/\|v_{i+1}\|_1$).*

2. *Given any $r$ and $q$,*

$$\mathrm{KL}(r,q) \geq \frac{\|r\|_1}{2}\mathrm{KL}(\tilde{r},\tilde{q}) + \mathbf{1}\left[\|q\|_1 \geq \|r\|_1\right]\frac{\|r\|_1}{2}\left(\sqrt{\frac{\|r\|_1}{\|q\|_1}} - \sqrt{\frac{\|q\|_1}{\|r\|_1}}\right)^2.$$

$$\geq \frac{\|r\|_1}{2}\mathrm{KL}(\tilde{r},\tilde{q}) + \mathbf{1}\left[\|q\|_1 \geq 2\|r\|_1\right]\frac{\|q\|_1}{8},$$

*where* $\mathrm{KL}(\tilde{r},\tilde{q}) \geq \frac{1}{2}\|\tilde{r}-\tilde{q}\|_1^2$.

*Proof.*   1. With $\phi(v) := \sum_s (v_s \ln v_s - v_s) = \langle v, \ln(v) - 1\rangle$, have

$$\nabla\phi(v) = \ln(v), \qquad (\nabla\phi)^{-1}(r) = \exp(r) = \nabla\phi^*(r),$$

and thus

$$D_\phi(a,b) = \langle a, \ln(a) - 1\rangle - \langle b, \ln(b) - 1\rangle - \langle \ln(b), a - b\rangle$$

$$= \left\langle a, \ln\left(\frac{a}{b}\right)\right\rangle - \langle 1, a - b\rangle.$$

By the equality case of Fenchel-Young inequality,

$$\phi^*(r) = \langle r, \nabla\phi^*(r)\rangle - \phi(\nabla\phi^*(r))$$

$$= \langle r, \exp(r)\rangle - \langle \exp(r), \ln(\exp(r)) - 1\rangle$$

$$= \langle 1, \exp(r)\rangle = \sum_s \exp(r_s).$$

Lastly, for any $v \geq 0$, by Jensen's inequality,

$$\min_{p\in\Delta_m} D(p,v) = \min_{p\in\Delta_m} \sum_s \left(p_s \ln(p_s/v_s) - p_s + v_s\right) = \|v\|_1 + \|v\|_1 \min_{p\in\Delta_m} \sum_s \frac{v_s}{\|v\|_1}\phi\left(\frac{p_s}{v_s}\right)$$

$$\geq \|v\|_1 + \|v\|_1 \min_{p\in\Delta_m} \phi\left(\sum_s \frac{p_s}{\|v\|_1}\right) = \ln\frac{1}{\|v\|_1},$$

which is attained uniquely with the choice $p = v/\|v\|_1$.

2. Define the scalar function $Q$ over $\mathbb{R}_{>0}$ as

$$Q(x) := \ln(x) + \frac{1}{x} - 1,$$

and rewrite KL with $Q$ as

$$\mathrm{KL}(r,q) = \left\langle r, \ln\frac{r}{q}\right\rangle + \langle 1, q - r\rangle$$

$$= \|r\|_1 \left\langle \tilde{r}, \ln\frac{\tilde{r}}{\tilde{q}}\right\rangle + \|r\|_1 \ln\frac{\|r\|_1}{\|q\|_1} + \langle 1, q - r\rangle$$

$$= \|r\|_1\mathrm{KL}(\tilde{r},\tilde{q}) + \|r\|_1 Q(\frac{\|r\|_1}{\|q\|_1}).$$

The first term is in the desired form, thus the remainder of the proof controls $Q$.

To start, here are some basic properties of $Q$. Since

$$Q'(x) = \frac{1}{x} - \frac{1}{x^2} = \frac{1}{x}\left(1 - \frac{1}{x}\right),$$

then $Q$ is decreasing along $(0, 1)$, attains a global minimum at 1, and increasing along $(1, \infty)$, specifically $Q(1) = 0$, meaning $Q \geq 0$ everywhere. Next,

$$Q''(x) = -\frac{1}{x^2} + \frac{2}{x^3} = \frac{1}{x^2}\left(\frac{2}{x} - 1\right),$$

whereby $Q$ is convex over $(0, 2)$ and concave over $[2, \infty)$; consequently, the remainder of the proof focuses on lower bounded $Q$ along $(0, 1]$, where it is convex, decreasing, and attains its global minimum at the right endpoint. To this end, for any $x \in (0, 1]$, note that

$$Q'(x) = -\int_x^1 Q''(r)\,\mathrm{d}r \leq -\int_x^1 \frac{1}{r^3}\,\mathrm{d}r = \frac{1}{2r^2}\Big|_x^1 = \frac{1}{2x^2} - \frac{1}{2},$$

$$Q(x) = -\int_x^1 Q'(r)\,\mathrm{d}r \geq -\frac{1}{2}\int_x^1 \left(\frac{1}{r^2} - 1\right)\mathrm{d}r = \frac{1}{2}\left(\frac{1}{r} + r\right)\Big|_x^1 = \frac{1}{2}\left(x + \frac{1}{x} - 2\right) = \frac{1}{2}\left(\sqrt{x} - \frac{1}{\sqrt{x}}\right)^2,$$

which by plugging in $x = \|r\|_1/\|q\|_1$ gives

$$Q\left(\frac{\|r\|_1}{\|q\|_1}\right) \geq \frac{1}{2}\mathbf{1}\left[\|q\|_1 \geq \|r\|_1\right]\left(\sqrt{\frac{\|r\|_1}{\|q\|_1}} - \sqrt{\frac{\|q\|_1}{\|r\|_1}}\right)^2$$

$$= \frac{\|q\|_1}{2\|r\|_1}\mathbf{1}\left[\|q\|_1 \geq \|r\|_1\right]\left(\frac{\|r\|_1}{\|q\|_1} - 1\right)^2$$

$$\geq \frac{\|q\|_1}{8\|r\|_1}\mathbf{1}\left[\|q\|_1 \geq 2\|r\|_1\right].$$

$\square$

Next, the reparameterization proof.

*Proof of Lemma 3.2.* First note that $|a_j(s)| = |b_j(s)|$ and $\mathrm{sgn}(a_j(s)b_j(s)) = \mathrm{sgn}(a_j(0)b_j(0)) = \beta_j$ for all $s$. Intuitively this follows from symmetry, but in detail, let $t > 0$ denote the earliest time when this claim fails, and notice

$$a_j(t) - a_j(0) = \int_0^t \frac{1}{n}\sum_i \ell'(\langle a(s) \odot b(s), F_i(s)\rangle)b_j(s)F_i(s)_j\,\mathrm{d}s$$

$$= \beta_j \int_0^t \frac{1}{n}\sum_i \ell'(\langle a(s) \odot b(s), F_i(s)\rangle)a_j(s)F_i(s)_j\,\mathrm{d}s$$

$$= \beta_j(b_j(t) - b_j(0)),$$

which means

$$a_j(t) - \beta_j b_j(t) = a_j(0) - \beta_j b_j(0) = 0$$

as desired; since $t$ was the assumed earliest violation, there is no violation and this equality holds for all $t$. These calculations also imply $a_j(t)^2 = b_j(t)^2 = \beta_j a_j(t)b_j(t)$ for all $t$.

For the second claim, again let $t$ denote the earliest violation, and it suffices to note that

$$\frac{\mathrm{d}}{\mathrm{d}s}\ln\left(\beta_j a_j(s)b_j(s)\right) = \frac{\beta_j\left[a_j(s)\dot{b}_j(s) + \dot{a}_j(s)b_j(s)\right]}{\beta_j a_j(s)b_j(s)}$$

$$= -\frac{\beta_j\frac{1}{n}\sum_i \ell'(\langle a \odot b, F_i\rangle)\left[a_j^2 + b_j^2\right]}{\beta_j a_j(s)b_j(s)}$$

$$= -2\nabla_u \mathcal{L}(\langle u \odot \beta, z_i\rangle) = \frac{\mathrm{d}}{\mathrm{d}s}\ln u,$$

and the proof completes by following the same steps as before, via the fundamental theorem of calculus. $\square$

Next, the basic implicitly-biased mirror flow guarantee.

*Proof of Lemma 3.3.* 1. Note

$$
\begin{aligned}
\frac{\mathrm{d}}{\mathrm{d}s} D_\phi(\bar{u}, u(s)) &= \frac{\mathrm{d}}{\mathrm{d}s} \left[ \phi(\bar{u}) - \phi(u(s)) - \left\langle \nabla\phi(u(s)), \bar{u} - u(s) \right\rangle \right] \\
&= -\left\langle \nabla\phi(u(s)), \frac{\mathrm{d}}{\mathrm{d}s} u(s) \right\rangle - \left\langle \frac{\mathrm{d}}{\mathrm{d}s}\nabla\phi(u(s)), \bar{u} - u(s) \right\rangle + \left\langle \nabla\phi(u(s)), \frac{\mathrm{d}}{\mathrm{d}s} u(s) \right\rangle \\
&= \left\langle \nabla f_s(u(s)), \bar{u} - u(s) \right\rangle \\
&\leq f_s(\bar{u}) - f_s(u(s)).
\end{aligned}
$$

By the fundamental theorem of calculus,

$$
D_\phi(\bar{u}, u(t)) - D_\phi(\bar{u}, u(0)) = \int_0^t \frac{\mathrm{d}}{\mathrm{d}s} D_\phi(\bar{u}, u(s)) \, \mathrm{d}s \leq \int_0^t \left( f_s(\bar{u}) - f_s(u(s)) \right) \mathrm{d}s. \quad (8)
$$

2. By eq. (8) and rearranging terms ,

$$
\int_0^t f_s(u(s)) \, \mathrm{d}s \leq D_\phi(\bar{u}, u(0)) - D_\phi(\bar{u}, u(t)) + \int_0^t f_s(\bar{u}) \, \mathrm{d}s. \quad (9)
$$

In addition, by part 2 of Lemma A.2 with $q = u_t$ and $r = \bar{u}$,

$$
\mathbf{1}\left[ \|u_t\|_1 \geq 2\|\bar{u}\|_1 \right] \frac{\|u_t\|_1}{8} \leq D_{\mathrm{KL}}(\bar{u}, u_t) - \frac{\|\bar{u}\|_1}{2} \mathrm{KL}\left( \frac{\bar{u}}{\|\bar{u}\|_1}, \frac{u_t}{\|u_t\|_1} \right) \leq D_{\mathrm{KL}}(\bar{u}, u_t).
$$

Adding the preceding inequality to the eq. (9) and using the identity $D_\phi(u, v) = \mathrm{KL}(u, v)$ and nonnegativity of $D_\phi$,

$$
\begin{aligned}
\mathbf{1}\left[ \|u_t\|_1 \geq 2\|\bar{u}\|_1 \right] \frac{\|u_t\|_1}{8} + \int_0^t f_s(u(s)) \, \mathrm{d}s &\leq D_\phi(\bar{u}, u(0)) - D_\phi(\bar{u}, u(t)) + \int_0^t f_s(\bar{u}) \, \mathrm{d}s \\
&\leq D_\phi(\bar{u}, u(0)) + \int_0^t f_s(\bar{u}) \, \mathrm{d}s \\
&= \mathrm{KL}(\bar{u}, u(0)) + \int_0^t f_s(\bar{u}) \, \mathrm{d}s. \quad (10)
\end{aligned}
$$

$\square$

Next comes the proof of the specialization to the logistic loss.

*Proof of Lemma 3.4.* Most of the bound is from the eq. (10); all that remains is the appearance of $G$ in the left hand side, and the appearance of 2 in the right hand side.

For the left hand side, note since $\ell \geq -\ell' \geq 0$ that

$$
\mathcal{L}_s(u) = \frac{1}{n} \sum_i \ell(\langle u, z_i(s) \rangle) \geq -\frac{1}{n} \sum_i \ell'(\langle u, z_i(s) \rangle) = G(s)
$$

For the right hand side,

$$
\begin{aligned}
\mathcal{L}_s(\bar{u}) &= \frac{1}{n} \sum_i \ln(1 + \exp(-\langle \bar{u}, z_i(s) \rangle)) \\
&\leq \frac{1}{n} \sum_i \exp(-\langle \bar{u}, z_i(s) \rangle) \\
&\leq \frac{1}{n} \sum_i \exp(-\ln(tn))
\end{aligned}
$$

$$\leq \frac{1}{tn}$$

and the bound follows by applying the preceding two inequalities on the left hand side and right hand side of eq. (10) respectively.

Instantiating the bound for every $s \leq t$ and using

$$\mathbf{1}\left[\|u_t\|_1 \geq 2\|\bar{u}\|_1\right]\frac{\|u_t\|_1}{8} \geq \frac{\|u_t\|_1 - 2\|\bar{u}\|_1}{8}$$

gives

$$\sup_{s \leq t}\left[\frac{\|u_t\|_1 - 2\|\bar{u}\|_1}{8} + 2\int_0^t G_s\,\mathrm{d}s\right] \leq R\|\bar{u}\|_1 + \frac{2}{n}.$$

In other words

$$\sup_{s \leq t}\|u_s\|_1 \leq (8R+2)\|\bar{u}\|_1 + \frac{16}{n}, \qquad 2\int_0^s G_r\,\mathrm{d}r \leq 2\int_0^s \mathcal{L}_r(u_r)\,\mathrm{d}r \leq R\|\bar{u}\|_1 + \frac{2}{n}.$$

Additionally, since $\exp(-\gamma\|\bar{u}\|_1) = \frac{1}{tn}$ and $\mathcal{L}_s(u_t) = \inf_{s \leq t}\mathcal{L}_s(u_s)$ as loss is monotonically decreasing, then

$$\mathcal{L}(u_t) \leq \frac{1}{t}\int_0^t \mathcal{L}_s(u_s)\,\mathrm{d}s \leq \frac{R\|\bar{u}\|_1}{2t} + \exp(-\gamma\|\bar{u}\|_1) = \exp(-\gamma\|\bar{u}\|_1)\left[\frac{nR\|\bar{u}\|_1}{2} + 1\right].$$

Since

The remaining items of the statement are established as follows.

1. If $t \geq t_1 := \left(\frac{20R}{\gamma}\right)^4$, then

$$R\|\bar{u}\|_1 + 2 = \left(R\ln(tn)\right)\gamma + 2 \leq \frac{10R(tn)^{1/4}}{\gamma} + 2 \leq \frac{20R(tn)^{1/4}}{\gamma} \leq \sqrt{tn},$$

   and therefore, defining $C_5 := \frac{R\|\bar{u}\|_1}{4} + 1$, then $\|\bar{u}\|_1 \geq \frac{2}{\gamma}\ln\frac{4}{n}C_5$, and thus, by Lemma A.1,

$$\min_i \frac{\langle z_i(t), u(t)\rangle}{\|u\|_1} \geq \frac{\gamma}{2(1+R)}.$$

2. Since $t \geq t_1$ and $\|u_t\|_1 \leq C\|\bar{u}\|_1$ with $C \leq 4 + R$, and by Lemma A.1 and using similar reasoning to the preceding,

$$\min_i \frac{\langle z_i(t), u(t)\rangle}{\|u\|_1} \geq \frac{\gamma}{2C}.$$

Lastly, since $u(s) > 0$ for all $s$ and $j$, assuming $\|z_i(s)\|_2 \leq 1$,

$$\ln u(t)_j - \ln u(0)_j = \int_0^t \frac{\mathrm{d}}{\mathrm{d}s}\ln u(s)_j\,\mathrm{d}s = \int_0^t \frac{1}{n}\sum_i \ell'(\langle u(s), z_i(s)\rangle)z_i(s)_j \leq \int_0^t G(s)\,\mathrm{d}s,$$

and thus

$$\|u(t)\|_1 = \sum_j u(t)_j \leq \sum_j u(0)_j \exp\left(\int_0^t G(s)\,\mathrm{d}s\right) = \|u(0)\|_1 \exp\left(\int_0^t G(s)\,\mathrm{d}s\right).$$

$\square$

To achieve only a constant-factor degradation in the margin requires a lot more chicanery, unfortunately. Firstly, these steps will in fact need the non-decreasing margin property, so it is a good time to prove it.

*Proof of Theorem 3.6.* It suffices to consider general $h$ as the special case of $h = \langle a \odot b, F \rangle$ follows since it is 2-homogeneous in $(a, b) \subseteq w_s$ and $\nabla_{(a,b)} h = F$ is bounded, which in turn implies the bounds with $u_s$ via the reparameterization in Lemma 3.3.

For the general $h$, let $\theta_s := w_s \setminus \hat{w}_s$ denote the denote potentially inhomogeneous parameters and define

$$\mathcal{L}_s(\hat{w}) := \frac{1}{n} \sum_i \ell(y_i h_i(x_i; \hat{w}, \theta_s)),$$

$$\psi_s(\hat{w}) := \ell^{-1}\left(n\mathcal{L}_s(\hat{w})\right).$$

To see that $\psi_s(\hat{w}_s)/\|\hat{w}_s\|^L$ is nondecreasing, note $\dot{w}_s = -\nabla \mathcal{L}_s$ and $\nabla \psi_s$ are colinear. Further, $\psi$ satisfies a homogeneity-like property even for the logistic loss: $\langle \partial_{\hat{w}} \psi_s(\hat{w}_s), \hat{w}_s \rangle \geq L\psi_s(w_s)$ (Ji & Telgarsky, 2020a, Lemma C.5). Abbreviating $\Psi_s := \psi_s(\hat{w}_s)$,

$$\dot{\Psi}_s := \langle \partial_w \psi_s(\hat{w}_s), \dot{w}_s \rangle$$
$$= \|\dot{w}_s\| \|\partial_w \psi_s(\hat{w}_s)\|$$
$$\geq \|\dot{\hat{w}}_s\| \|\partial_{\hat{w}} \psi_s(\hat{w}_s)\|$$
$$\geq \|\dot{\hat{w}}_s\| \left\langle \partial_{\hat{w}} \psi_s(\hat{w}_s), \frac{\hat{w}_s}{\|\hat{w}_s\|} \right\rangle$$
$$\geq \frac{\|\dot{\hat{w}}_s\|}{\|\hat{w}_s\|} L\Psi_s. \tag{11}$$

Denoting $r_s := \|\hat{w}_s\|^L$,

$$\dot{r}_s = \frac{L}{2} \langle \hat{w}_s, \hat{w}_s \rangle^{L/2-1} 2 \langle w_s, \dot{\hat{w}}_s \rangle \tag{12}$$
$$= L \frac{\|\hat{w}_s\|^L}{\|\hat{w}_s\|} \left\langle \frac{\hat{w}_s}{\|\hat{w}_s\|}, \dot{\hat{w}}_s \right\rangle \leq Lr_s \frac{\|\dot{\hat{w}}_s\|}{\|\hat{w}_s\|}. \tag{13}$$

The second equality in eq. (11) implies $\dot{\Psi}_s \geq 0$ and hence the unnormalized smoothed margins are nondecreasing. In particular, $\Psi_s \geq \Psi_0 > 0$. Therefore, by quotient rule and eqs. (11) and (12),

$$\frac{d}{dt} \frac{\Psi_s}{\|\hat{w}_s\|^L} = \frac{d}{ds} \frac{\Psi_s}{r_s} = \frac{\dot{\Psi}_s r_s - \Psi_s \dot{r}_s}{r_s^2} \geq 0.$$

To see that norms are increasing, first recall from above $\Psi_s > 0$. Since $-\ell' \in (0, 1)$,

$$\frac{d}{ds} \ln \|\hat{w}_s\|_2^2 = \frac{2}{\|\hat{w}_s\|^2} \langle \hat{w}_s, \dot{\hat{w}}_s \rangle \tag{14}$$
$$= \frac{2}{\|\hat{w}_s\|^2} \langle \hat{w}_s, \partial_{\hat{w}} \psi_s(\hat{w}_s) \rangle \cdot \left(-\ell'(\Psi_s)\right) \tag{15}$$
$$\geq \frac{2L}{\|\hat{w}_s\|^2} \Psi_s \cdot \left(-\ell'(\Psi_s)\right) \tag{16}$$
$$> 0. \tag{17}$$

This is sufficient to show that $\|\hat{w}_s\| \to \infty$. Suppose otherwise. It suffices to show that $\frac{d}{ds} \ln \|\hat{w}_s\|_2^2$ is bounded below by a positive constant since that implies $\|\hat{w}_s\| \to \infty$ which is a contradiction. First note that loss $\mathcal{L}_s(\hat{w}_s)$ can be bounded from below by positive constant. To see this, since $\|\hat{w}_s\| \not\to \infty$ but $\frac{d}{ds} \ln \|\hat{w}_s\|_2^2 > 0$ by eq. (14), it follows that $B := \sup_s \|\hat{w}_s\| < \infty$. Recall $C := \sup_\theta \sup_{\|\hat{w}\| \leq 1} \|\partial_{\hat{w}} h(x_i; \hat{w}, \theta)\| < \infty$ by hypothesis. Hence, by homogeneity of $h$ and $\partial_{\hat{w}} h$, and since $\ell$ is nonincreasing,

$$n\mathcal{L}_s(\hat{w}_s) = \sum_i \ell(y_i h(x_i; \hat{w}_s, \theta_s))$$
$$= \sum_i \ell(\frac{y_i}{L} \langle \partial_{\hat{w}} h(x_i; \hat{w}_s, \theta_s), \hat{w}_s \rangle)$$

$$\geq \sum_i \ell(\frac{y_i}{L} \|\partial_{\hat{w}} h(x_i; \hat{w}_s, \theta_s)\| \|\hat{w}_s\|)$$

$$= \sum_i \ell(\frac{1}{L} \left\|\partial_{\hat{w}} h(x_i; \frac{\hat{w}_s}{\|\hat{w}_s\|}, \theta_s)\right\| \|\hat{w}_s\|^L)$$

$$\geq \sum_i \ell(\frac{1}{L} C B^L)$$

$$= n\ell(\frac{CB^L}{L})$$

This in turn implies $\Psi_s := \ell^{-1}(n\mathcal{L}_s(\hat{w}_s))$ and $-\ell'(\Psi_s)$ are bounded below by positive constants. Hence, eq. (14) implies $\frac{\mathrm{d}}{\mathrm{d}s} \ln \|\hat{w}_s\|_2^2$ is bounded below by a positive constant.

$\square$

With that out of the way, now comes a messier lemma: consider the time-rescaled flow on the smooth margin directly. Controlling for the time-rescaling leads to the doubly-exponential sufficient condition on $t$. This proof invokes Theorem 3.6.

**Lemma A.3.** *Suppose $t \geq t_1$ as defined in Lemma 3.4 and there exists $\bar{p} \geq 0$ and $\gamma > 0$ with $\|\bar{p}\|_1 = 1$ and $\inf_{s \in [0,t]} \min_i \langle \bar{p}, z_i(s) \rangle \geq \gamma$. Then the smoothed margin satisfies,*

$$\frac{\psi_t(u(t))}{\|u(t)\|_1} \geq \gamma - \frac{1 + 2\ln n}{\|u_t\|_1} - \frac{D_\phi(\bar{u}, u(0))}{\|u_t\|_1 \xi_t}.$$

*Proof.* By the second equality in eq. (11), it follows that the *unnormalized* smoothed margin $\psi_s(u(s))$ is nondecreasing. Hence, $\psi(u(t)) = \sup_{s \in [0,t]} \psi(u(s))$. By the equality case of the MF bound in Lemma 3.3 applied to $\mathcal{L}_s$ and defining $\bar{u} = \|u_t\|_1 \bar{p}$,

$$D_\phi(\bar{u}, u(t)) - D_\phi(\bar{u}, u(0)) = \int_0^t \langle \nabla f_s(u(s)), \bar{u} - u(s) \rangle \, \mathrm{d}s$$

$$= 2 \int_0^t \langle \nabla \mathcal{L}_s(u(s)), \bar{u} - u(s) \rangle \, \mathrm{d}s$$

$$= 2 \int_0^t \langle -\nabla \psi_s(u(s)), \bar{u} - u(s) \rangle \left( \frac{-\ell'(\psi_s(u_s))}{n} \right) \mathrm{d}s$$

$$\leq 2 \int_0^t \left[ -\psi_s(\bar{u}) + \psi_s(u_s) \right] \left( \frac{-\ell'(\psi_s(u_s))}{n} \right) \mathrm{d}s$$

$$\leq 2 \left[ \psi_t(u(t)) - \gamma\|\bar{u}\|_1 + 2\ln n + 1 \right] \int_0^t \frac{-\ell'(\psi_s(u_s))}{n} \, \mathrm{d}s.$$

Defining $\xi_t := -\frac{1}{n} \int_{t_1}^t \ell'(\psi_s(u_s)) \, \mathrm{d}s$ and dividing both sides by $2\|u_t\|_1 \xi_t$, rearranging terms, and recalling that $D_\phi \geq 0$,

$$\frac{\psi_t(u_t)}{\|u_t\|_1} \geq \gamma - \frac{1 + 2\ln n}{\|u_t\|_1} + \frac{D_\phi(\bar{u}, u(t)) - D_\phi(\bar{u}, u(0))}{\|u_t\|_1 \xi_t}$$

$$\geq \gamma - \frac{1 + 2\ln n}{\|u_t\|_1} - \frac{D_\phi(\bar{u}, u(0))}{\|u_t\|_1 \xi_t}. \tag{18}$$

$\square$

Combining these pieces leads to the general (slow) margin guarantee.

**Lemma A.4.** *If $t \geq e + \exp(\exp(4nK(\bar{p}, u(0)))/\gamma))$ and $\min_{s \leq t} \min_i \bar{p}^\mathsf{T} z_i(s) \geq \gamma$, then*

$$\min_i \frac{\langle u(t), z_i(t) \rangle}{\|u(t)\|_1} \geq \frac{\gamma}{16}.$$

*Proof.* Define $\bar{u} := \ln(tn)\bar{p}/\gamma$ as in Lemma 3.4, and consider two cases.

- If $\|u_t\|_1 \le 8\|\bar{u}\|_1$, then by the second part of Lemma 3.4,

$$\min_i \frac{\langle u(t), z_i(t) \rangle}{\|u(t)\|_1} \ge \frac{\gamma}{16}.$$

- Otherwise, $\|u_t\|_1 > 8\|\bar{u}\|_1 \ge 8\ln(tn)/\gamma$, and so by Lemma A.3 and Lemma A.1,

$$\min_i \frac{\langle u(t), z_i(t) \rangle}{\|u(t)\|_1} \ge \frac{\psi_t(u_t)}{\|u_t\|_1} \ge \gamma - \frac{\gamma(1 + 2\ln n)}{8\ln(tn)}$$

In particular, if $t \ge \max\{e, \exp(\exp(4nK(\bar{p}, u(0))/\gamma))\}$, then this simplifies to

$$\frac{\psi_t(u_t)}{\|u_t\|_1} \ge \gamma - \frac{\gamma}{4} - \frac{\gamma}{4} \ge \frac{\gamma}{2}.$$

$\square$

*Proof of Theorem 3.1.* The proof is a combination of the preceding pieces and the following chain rule calculation. Recalling the definition of $C_1$ and $C_2$ and and making use of the final bounds in the unconditional part of Lemma 3.4,

$$\left| \langle \bar{p}, z_i(\tau) - z_i(0) \rangle \right| \le \sum_j \bar{p}_j \int_0^\tau \left| \dot{z}_{i,j}(s) \right| \mathrm{d}s$$

$$= \sum_j \bar{p}_j \int_0^\tau \left| \nabla_V z_{i,j}(s) \dot{V} \right| \mathrm{d}s$$

$$= \sum_j \bar{p}_j \int_0^\tau \left| \nabla_V z_{i,j}(s) \sum_r \ell'_r \sum_k u_k \nabla_V z_{r,k} \right| \mathrm{d}s$$

$$\le \sum_j \bar{p}_j \int_0^\tau \sum_r |\ell'_r| \left[ u_j C_1 + C_2 \sum_k u_k \right] \mathrm{d}s$$

$$\le \left( C_1 \|\bar{p}\|_\infty + C_2 \|\bar{p}\|_1 \right) \left[ \int_0^\tau G(s)\,\mathrm{d}s \right] \left( \sup_{s \le \tau} \|u(s)\|_1 \right)$$

$$\le \left( C_1 \|\bar{p}\|_\infty + C_2 \right) \left[ \|\bar{u}\|_1 (8R + 2) + 32 \right]^2.$$

Then if

$$\|\bar{p}\|_\infty \le \frac{\gamma}{4C_1 \left[ \|\bar{u}\|_1 (8R + 2) + 32 \right]^2}, \qquad C_2 \le \frac{\gamma}{4 \left[ \|\bar{u}\|_1 (8R + 2) + 32 \right]^2},$$

for any $s \le t$,

$$\langle \bar{p}, z_i(s) \rangle \ge \langle \bar{p}, z_i(0) \rangle - \left| \langle \bar{p}, z_i(s) - z_i(0) \rangle \right| \ge \frac{\gamma}{2}.$$

Thus, applying first part of lemma 3.4 and lemma A.4 gives the desired margin lower bounds. $\square$

*Proof of Theorem 1.1.* It suffices to combine Lemma 3.2 and Theorem 3.1 and Theorem 3.6. $\square$

# B   DEFERRED PROOFS FROM SECTION 4

To start, the sampling guarantee.

**Lemma B.1.** *Let examples $((x_i, y_i))_{i=1}^n$ be given, and suppose $\gamma_{\mathcal{D}} > 0$. Let width $m$ be given with*

$$m \geq m_0 := 2^{10} \ln(2n/(\delta\gamma))^2 \left(\frac{8}{\gamma}\right)^{2d-2},$$

*and let $(v_j)_{j=1}^m$ be uniformly IID over the surface of the sphere and $(\beta_j)_{j=1}^m$ uniformly from $\{\pm 1\}^m$, and define $z_{i,j} := y_i \beta_j \sigma(v_j^\mathsf{T} x_i)$. With probability at least $1 - \delta$, there exists a discrete unnormalized nonnegative measure $(u_j)_{j=1}^m$ with*

$$\min_i \langle u, z_i \rangle \geq \frac{\bar{\gamma}}{4},$$

$$D_\phi\left(u, (1/m, \ldots, 1/m)\right) \leq (d-1)\ln\frac{48}{\gamma},$$

$$\left|(a_1^2, \ldots, a_m^2)\right|_\infty \leq \frac{1}{\sqrt{m}}.$$

*Proof.* Take any measure $\mu$ which achieves margin $3\bar{\gamma}/4$ in the definition of $\bar{\gamma}$; the proof first discretizes this measure, and then produces a final measure which uses random weights on the sphere.

By the Maurey sampling method (Telgarsky, 2024, Lemma 3.2), there exists a discrete unnormalized measure $\mu^{(1)}$ with $k$ atoms there exist $((\mu_j^{(1)}, \beta_j^{(1)}, v_j^{(1)}))_{j=1}^k$ with $\mu_j^{(1)} = 1/\sqrt{k}$, $\beta_j^{(1)} \in \{\pm 1\}$, and $\|v_j^{(1)}\|_2 = 1$, and a corresponding $z_{i,j}^{(1)} := y_i \beta_j^{(1)} \sigma(x_i^\mathsf{T} v_j^{(1)})$ such that

$$\frac{1}{n} \max_i \left(\left\langle \mu^{(1)}, z_i^{(1)} \right\rangle - y_i \int \sigma(v^\mathsf{T} x_i)\, d\mu(v)\right)^2 \leq \frac{1}{n} \sum_i \left(\left\langle \mu^{(1)}, z_i^{(1)} \right\rangle - y_i \int \sigma(v^\mathsf{T} x_i)\, d\mu(v)\right)^2 \leq \frac{1}{k};$$

in other words, to ensure $\mu^{(1)}$ has margin at least $\gamma/2$, it suffices to take $k = 16n/\gamma^2$. The remainder of the proof adjusts for the fact that $(\beta_j^{(1)}, v_j^{(1)})$ are not distributed uniformly.

To finish, produce a final measure $(u_j)_{j=1}^m$ as follows. For each of the $k$ atoms forming $\mu^{(1)}$, sample $m/k$ pairs $(\beta, v)$ from the product distribution which is uniform on $\beta \in \{\pm 1\}$ and has $v$ uniform on the sphere. By standard results about sampling on the sphere (Ball, 1997, Lemma 2.3), the probability that a sampled $v$ satisfies $\|v - v_j\| \leq \epsilon$ is at least $\frac{1}{2}(\epsilon/2)^{d-1}$, thus the probability of this event occurring and that the corresponding sign $\beta_j$ matches $\beta_j^{(1)}$ is at least $\tau := \frac{1}{4}(\epsilon/2)^{d-1}$. By Hoeffding's inequality, with probability at least $1 - \delta_1$, the exact number $m\hat{\tau}_j$ of sampled pairs satisfying this condition across all $j$ satisfies

$$\max_j \left| m\hat{\tau}_j - m\tau \right| \leq \sqrt{\frac{m\ln(2k/\delta_1)}{2k}}.$$

As such, by choosing $\delta_1 = \delta/(2k)$ and the choice $m \geq m_0$ and $\epsilon = \gamma/4$, it follows that the measure $(u_j)_{j=1}^m$ obtained by assigning $u_j = 1/(m\hat{\tau}_j)$ to the $m\hat{\tau}_j$ triples satisfying the corresponding event above (where $v_{(j)}$ is the weight in $\mu^{(1)}$ associated with $v_j$, and letting $\mu_0$ denote the uniform measure $(1/m, \ldots, 1/m)$)

$$\max_i \left| \langle u, z_i \rangle - \left\langle \mu^{(1)}, z_i^{(1)} \right\rangle \right| \leq \max_j \|\sigma(v_j^\mathsf{T} x) - \sigma(v_{(j)}^\mathsf{T} x)\| \leq \max_j \|v_j - v_{(j)}\| \leq \frac{\gamma}{4},$$

$$\max_j \left| m\hat{\tau}_j - m\tau \right| \leq \sqrt{\frac{m\ln(2k/\delta_1)}{2k}} = m\sqrt{\frac{\ln(2k/\delta_1)}{2mk}} \leq \frac{m\tau}{2},$$

$$D_\phi(u, \mu_0) = \sum_j u_j \ln\frac{u_j}{(\mu_0)_j} = \sum_j \frac{m\hat{\tau}_j}{m\hat{\tau}_j} \ln\frac{1/(m\hat{\tau}_j)}{1/m} \leq \ln\frac{3}{2\tau} \leq (d-1)\ln\frac{48}{\gamma},$$

$$\|u\|_\infty \leq \frac{3}{2m\tau} \leq \frac{1}{\sqrt{m}},$$

since

$$m \geq \left(\frac{2\ln(2k/\delta)}{\tau}\right)^2 \geq 2^{10}\ln(2n/(\delta\gamma))^2 \left(\frac{8}{\gamma}\right)^{2d-2}.$$

$\square$

*Proof of Theorem 4.1.* The proof follows by combining the preceding sampling guarantee in Lemma B.1 with Theorem 3.1. Furthermore, the margin guarantees are converted into test error guarantees via a margin based Rademacher complexity generalization bound. Recall the definition of $C_1, C_2$.

$$C_1 := \sup_{s<t} \sup_{i,r,j} \left| \langle \nabla_V F_j(x_i; V(s)), \nabla_V F_j(x_r; V(s)) \rangle \right|,$$

$$C_2 := \sup_{s<t} \sup_{i,r,j\neq k} \left| \langle \nabla_V F_j(x_i; V(s)), \nabla_V F_k(x_r; V(s)) \rangle \right|,$$

To bound $C_1$ note that for any inputs $x, x' \in \mathbb{R}^d$, index $j \in [m]$, and $V \in \mathbb{R}^{m\times d}$,

$$\left| \langle \nabla_V F_j(x; V), \nabla_V F_j(x', V) \rangle \right| = \left| \left\langle \sigma'(\tilde{v}_j x) e_j x^{\mathsf{T}} \left( I - \tilde{v}_j \tilde{v}_j^{\mathsf{T}} \right), \sigma'(\tilde{v}_j x') e_j x'^{\mathsf{T}} \left( I - \tilde{v}_j \tilde{v}_j^{\mathsf{T}} \right) \right\rangle \right|$$

$$\leq \left\| \left( I - \tilde{v}_j \tilde{v}_j^{\mathsf{T}} \right) x \right\| \left\| \left( I - \tilde{v}_j \tilde{v}_j^{\mathsf{T}} \right) x' \right\|$$

$$\leq \|x\| \|x'\|$$

$$\leq 1.$$

Hence, $C_1 \leq 1$. By similar calculations and noting that $e_j \mathsf{T} e_k = 0$ for $j \neq k$ implies $C_2 = 0$.

Now by lemma B.1 and theorem 3.1 and above calculations for $C_1, C_2$, grants the following margin bounds.

If $t \geq \left( \frac{20R}{\gamma} \right)^4$,

$$\min_i \frac{y_i h(x_i; w)}{\|a(t)\|^2 + \|b(t)\|^2} \geq \frac{\gamma}{4(1+R)}.$$

If additionally $t \geq e + \exp(\exp(4nR/\gamma))$, then

$$\min_i \frac{y_i h(x_i; w)}{\|a(t)\|^2 + \|b(t)\|^2} \geq \frac{\gamma}{32}.$$

Let us now establish the test error guarantee. Take margin $\gamma$ The following argument uses the same proof scheme in Telgarsky (2022, proof of lemma 2.5). Denote the symmetric convex hull of a set $S$ as

$$\mathrm{sconv}(S) := \left\{ \sum_{j\in[m]} p_j s_j \ : \ m \geq 0, \ p \in \mathbb{R}^m, \|p\|_1 \leq 1, \ s_j \in S \right\}.$$

Now consider the hypothesis class,

$$\mathcal{F} := \left\{ x \to \frac{1}{\|u\|_1} \sum_{j=1}^m u_j \beta_j \sigma \left( \langle v_j/\|v_j\|, x \rangle \right) \ : \ m \geq 0 \, u \geq 0, \ u \in \mathbb{R}^m, \ \beta_j \in \{\pm 1\}, \ v_j \in \mathbb{R}^d \right\}$$

$$\subset \left\{ x \to \sum_{j=1}^m p_j \sigma \left( \langle s_j, x \rangle \right) \ : \ m \geq 0 \, p \in \mathbb{R}^m, \|p\|_1 \leq 1, \|s_j\| = 1 \, \beta_j \in \{\pm 1\}, \ v_j \in \mathbb{R}^d \right\}$$

$$= \mathrm{sconv} \left( \{ x \to \sigma(\langle v, x \rangle) \} \ : \ \|v\|_2 = 1 \right).$$

By standard Rademacher calculus (Shalev-Shwartz & Ben-David, 2014),

$$\mathrm{Rad}(\mathcal{F}) \leq \mathrm{Rad}(\mathrm{sconv} \left( \{ x \to \sigma(\langle v, x \rangle) \} \ : \ \|v\|_2 = 1 \right)) \leq \frac{1}{\sqrt{n}},$$

and thus, by the margin bound for Rademacher complexity (Srebro et al., 2010, Theorem 5), with probability at least $1 - \delta$, simultaneously for all $\gamma > 0$ and all width $m$, every $w = (a, b, V)$ with

$$\min_{i\in[n]} \frac{y_i h(x_i; w)}{\|a\|^2 + \|b\|^2} \geq \gamma$$

satisfies

$$\Pr\left[yh(x;w) \leq 0\right] = \mathcal{O}\left(\frac{\ln(n)^3}{\gamma^2}\mathrm{Rad}(\mathcal{F})^2 + \frac{\ln\ln\frac{1}{\gamma} + \ln\frac{1}{\delta}}{n}\right) = \mathcal{O}\left(\frac{\ln(n)^3}{n\gamma^2} + \frac{\ln\frac{1}{\delta}}{n}\right).$$

Applying the margin based Rademacher generalization bound for margin obtained for $t \geq t_1$ and margin obtained for $t \geq e + \exp(\exp(4nR/\gamma))$ gives the corresponding test error bounds.

$\square$

