# OpenReview forum: "The ubiquity of 2-homogeneity, how its implicit bias selects features, and other stories"
_ICLR.cc/2025/Conference — Submitted to ICLR 2025_

### Official Review · Reviewer_GWqR · 2024-10-20

**Soundness:** 4
**Presentation:** 3
**Contribution:** 3
**Rating:** 6
**Confidence:** 3

**Summary:**

The paper studies the gradient flow dynamics on networks where the last layer includes normalization followed by a linear transformation. Formally, it means that they consider networks where the last layers are a two-layer linear diagonal network, but the motivation for studying this architecture comes from normalization layers. They show implicit bias towards ell_1 margin maximization of the linear transformation induced by the last layer (with its normalization). This result includes abstract assumptions, but they also consider two-layer normalized ReLU networks as a concrete instantiation, obtain generalization guarantees, and use it to achieve a generalization guarantee for sparse parities which goes beyond the NTK bounds.

**Strengths:**

The paper considers the challenging problem of analyzing the implicit bias of non-homogenous networks. They give a general margin-maximization guarantee (under certain assumptions) and show that this general result can be used to obtain more concrete bounds for two-layer normalized ReLU networks and specifically for learning sparse parities. Unlike many existing results on the implicit bias in the rich regime, their result gives a bound on the margin rather than just showing convergence to a KKT point of the max-margin problem. The sparse parities learning problem has attracted much interest, and several prior works considered learning sparse parities using shallow neural networks, and thus the corollary (Example 4.1) is of independent interest. The technique seems interesting and non-trivial, and might also be of independent interest.

**Weaknesses:**

Theorem 3.1 is very hard to parse. It includes assumptions that involve the margin induced by the random features at initialization, and the KL divergence between the ell_1 max margin predictor at initialization and the last layer’s initialization. It also includes an assumption called “the slow growing derivatives property” which might be hard to verify. So, it’s hard for me to determine to what extent Theorem 3.1 can be applied in different settings. The good news is that the author/authors prove that it can be applied to sufficiently wide two-layer normalized ReLU networks.

Typos:
- Title is missing
- Line 433: xi_t —> \xi_t
- Line 479: “of of”

**Questions:**

Can you compare your result on two-layer networks to Theorem 3.3 in Telgarsky 2022?

---

> ### Author Response · Authors · 2024-11-27
> **Rebuttal by Authors**
>
> We thank the reviewer for their time and comments. We address the reviewer’s comments below.
>
> > Can you compare your result on two-layer networks to Theorem 3.3 in Telgarsky 2022?
>
> The primary difference is that theorem 3.3 in Telgarsky 2022 requires the inner
> layer to be frozen while our theorem 1.1 does not. In particular, it is not
> clear how to adapt the potential argument used in theorem 3.3 in
> Telgarsky 2022 to handle rotation.

---

> > ### Comment · Reviewer_GWqR · 2024-12-02
> >
> > Thanks for your response. I will keep my positive recommendation.

---

### Official Review · Reviewer_XxTj · 2024-11-04

**Soundness:** 3
**Presentation:** 3
**Contribution:** 2
**Rating:** 6
**Confidence:** 3

**Summary:**

The paper describes the dynamics of a network with a 2-homogeneous head and establish the margin guarantee for the models in late training phases. Under a simpler setting of adding 2-homogeneous head to a one-hidden-layer network, the paper can further establish generalization guarantees based on the margin guarantees.

**Strengths:**

The paper establishes theorem 3.6, a generalization of result from previous work in a clear form. The paper has clear writing that exhibits the proof ideas.

**Weaknesses:**

1. Theorem 3.1 & 4.1 has assumptions that are hard to verify, and I believe these assumptions should be given out more explicitly in the texts. For theorem 3.1, as the line 260 RHS bounds depends on $t$, the choice of $t$ depends on $R$ and $n$, and the LHS quantity also depends on $n, R$ and $t$, it is essentially hard to tell whether these assumptions can be satisfied. For theorem 4.1, $m$ depends on $t$, $t$ depends on $R$ and $R$ may depend on $m$ as $m$ changes the initialization scheme. Thus the authors should at least provide the scale estimate of $R$ to show that their assumption is reasonable.

2. The proof technique for Theorem 3.6 is standard and may be lacking of novelty compared to former work (https://arxiv.org/abs/1906.05890) .

3. The correlation of part1 and part2 of theorem 1.1 are unclear from text. It seems like two independent results. The main claim of the current paper - selecting feature from space $F_1$ is more efficient than from $F_2$ using 2-homogeneous head - does not require a result in representation learning, thus is basically already inferred from previous work (Telgarsky 2022 + Lyu 2019, as mentioned in Example 4.1). Also it is unclear from text why 2-homogeneity is unique - from my perspective $k$-homogeneity may also be provable using similar arguments for $k>1$.

Furthermore, the title of the paper should be changed in the pdf file.

**Questions:**

Regarding weakness 1, can the authors provide an instance of choices of the parameters such that the assumptions in theorem 3.1&4.1 are satisfied? And please provide more clarifications on the strength of these assumptions?

Also can the author further clarify weakness 2 & 3?

---

> ### Author Response · Authors · 2024-11-27
> **Rebuttal by Authors**
>
> We thank the reviewer for their time and comments. We address the reviewer’s
> comments below.
>
> 1.  We first provide parameter choices for theorem 3.1
> Without a distributional margin assumption like the one stated in section 4,
> $R \leq \ln (m)$. Therefore, setting $ t = O {( \frac{ \ln (n) }{ \gamma} )}^4 $ for the first
> margin result and $t = O( \exp ( m^{(4 n / \gamma)})) $ suffices.
> For theorem 4.1, thanks to the margin distribution assumption and sampling tools used
> in the proof of theorem 4.1, one can establish $R \leq O ( d \ln (1 / \gamma) )$. Hence,
> $t = ( {(\frac { d \ln (1 / \gamma )}{ \gamma} )}^4)$ suffices. It should be noted that all the assumptions for theorem 3.1 are satisfied for shallow ReLU networks provided the distributional margin assumption in section 4 holds. The distributional margin assumption is rather mild as it holds as long as there exists a perfect classifier.
>
>
>
> 2. As we will elaborate shortly, the proof technique of Theorem 3.6 is distinct from the prior work of Lyu&Li, which grants a more general theorem that applies to various classes of inhomogeneous functions such as ResNet and transformer models. In particular, Lyu&Li analyzes the logarithm of the smoothed
> margin by decomposing into its radial and tangential components. It is not clear how to
> use the radial and tangential decomposition argument when there exists
> inhomogeneous parameters. To avoid using this decomposition, we deal directly with
> the smoothed margin which allows us to establish a more general theorem.
> Furthermore, part 1 of Theorem 1.1 establishes that GF obtains a large margin while
> Lyu&Li assume one starts with a positive initial margin.  We have revised the section following theorem 3.6 to include the
> discussion comparing theorem 3.6 and lemma 5.2 in Lyu&Li.
>
> 3. Thank you for raising this comment. Part 1 and part 2 both characterize
> the implicit bias of gradient descent; specifically, gradient flow
> implicitly maximizes the margin of the model with respect to the variational norm
> or, equivalently, optimizing over the $F_1$ space. Part 1 establishes that after some training, gradient flow obtains large $F_1$ margin. Part 2 ensures that the $F_1$ margin
> of the model does not degrade over time and hence gradient flow continues to implicitly
> maximize the $F_1$ margin.
>
> Additionally, the combination of Telgarsky and Lyu&Li
> is not enough to establish theorem 1.1. In particular, when trying to establish part 1, while theorem 3.3 in Telgarsky does achieve the $F_1$ margin, the work freezes the inner layer which vastly simplifies
> the training dynamics. On the other hand, while theorem 2.2 in Telgarsky does not freeze
> the inner layer, it achieves the worse NTK margin. In particular, for the 2 parity problem,
> the $F_1$ margin is $O( \frac{ 1 }{ \sqrt{d }})$ which the NTK margin is $ O(\frac 1 d) $
> which is a significant difference; in the context of margin based generalization bounds, if a model only achieves the NTK margin, to get the same generalization error of
> a model that achieves the $F_1$ margin, the former model requires $O(d)$ more samples.
>
> > Also it is unclear from text why 2-homogeneity is unique - from my
> > perspective $k$-homogeneity may also be
> > provable using similar arguments for $k>1$.
>
> The power of 2-homogeneity is best demonstrated by lemma 3.2.
> Given a more general $k$ homogeneous model of the form
> $h(x ; w = (a^1,\dots a^L ,V)) = \langle a^1 \odot \dots \odot a^L,  F(x; V) \rangle$ and
> defining the reparameterized weights as $u_j(t) := \prod_{\ell = 1}^k a^{\ell}_j(t)$,
> one can, by calculation, determine that the reparameterized mirror flow (eq 3) fails
> to hold for $k \neq 2$. As discussed in the section following lemma 3.2, the
> reparameterized mirror flow explicitly optimizes over the superior $F_1$ space
> as opposed to $F_2$ space. This is extremely important as $F_2$ margin can be significantly worse than the $F_1$ margin. In particular, as mentioned in example 4.1, for the k- parity problem $F_1$ margin is $ \frac { 1 }{ k \sqrt{d }}$ while the
> $F_2$ is $ \frac { 1 }{  d^{ k /2} }$; the significance of this difference
> becomes apparent again when looking at margin based generalization bounds as models that
> achieve $F_2$ margin will require exponentially more samples in $d$ to achieve the same
> generalization error as a model that achieves $F_1$ margin.
> To provide additional clarity, we have incorporated the preceding discussion in the section following lemma 3.2
> in order to highlight the importance of 2-homogeneity.
>
> >Furthermore, the title of the paper should be changed in the pdf file.
>
> Thank you for pointing this out. We have updated the title.
>
> If the reviewer is not satisfied with this response, can they please clarify their further concerns?

---

> > ### Comment · Reviewer_XxTj · 2024-12-01
> >
> > Thank you for the clarification. I guess the most crucial concern is weakness No.1, which is still confusing as the response does not talk about the implicit assumptions for theorem 3.1 (line 261). I suggest to make the following clarifications in the texts:
> > 1. the bound t ~ e^{e^{4nT/\gamma}} is doubly exponentially large - thus plugging into the constraints of theorem 3.1, we should have $\Vert{\bar{p}}_0$ and $C_2$ to be exponentially small - which is actually questionable as assumptions. I suggest giving an example that the scale of the quantities actually work (e.g. 2-layer network).
> > 2. The discussion over the scale of $C_1$ and $C_2$ is very non-rigorous - i.e. for the paragraph starting from 274, it is unclear how $C_1$ is related to the network width and is also unclear for the NTK example. As the quantities $C_1, C_2$ are very crucial to the soundness of the theorem and are hard to evaluate, I strongly suggest make rigorous illustrations for this discussion.
> > 3. Another important thing is for $\bar{p}$. $\bar{p}$ should correspond to a large margin solution for the network, thus the constraint for $\Vert\bar{p}\Vert_0$ also is an implicit assumption on the data distribution. Thus it is also unclear if this assumption can actually be realized.

---

> > > ### Author Response · Authors · 2024-12-04
> > >
> > > Thank you for your time and for clarifying!
> > > 1. We first address the concern regarding $C_2$.
> > > Working with 2 layer ReLU networks, for any
> > > inputs $x , x' \in \mathcal{R}^d$, indices $j,k \in [m]$ such that $j \neq k$, and $V \in \mathcal{R}^{m \times d}$,
> > > since $e_j^\intercal e_k = 0$,
> > > $$
> > >  \left | \langle \nabla_V F_j(x ; V), \nabla_V F_k(x',V) \rangle \right |
> > >  = \left | \langle \sigma'( \tilde { v}_j x)
> > >  e_j x^\intercal ( I - \tilde { v}_j \tilde { v}_j^\intercal ), \sigma'( \tilde { v}_j x')
> > >  e_k x'^\intercal ( I - \tilde { v }_j \tilde { v }_j^\intercal ) \rangle \right |
> > >  = 0.
> > > $$
> > > From the preceding calculation and definition of $C_2$, it follows that $C_2 = 0$.
> > > We now address the concern regarding $\bar{p}$.
> > > By the distributional margin assumption (line 500), we can *construct*
> > > $\bar{p}$ via sampling methods (for a complete proof, we refer the reviewer to lemma B.1) such that its norm satisfies
> > > $\\| \bar{p} \\|\_{\infty}  \leq \frac 1 {\sqrt{m}} $. Hence, $\\| \bar{p} \\|\_{\infty}$ can be made exponentially small by making the width $m$ exponentially large.
> > >
> > > For the sake of clarity, we now provide the scale of every relevant quantity in theorem 3.1 for shallow networks. As discussed above, $C_2 = 0$ and, by lines
> > > 1251 - 1261 in the appendix, $C_1 = 1$. Now thanks to the distributional margin assumption (line 500), we invoke lemma B.1 to construct $\bar{p}$
> > > such that $\\| \bar{p} \\|\_{\infty} \leq \frac 1 {\sqrt{m}}$ and
> > > $R := KL( \bar{p}, u(0)) \leq d \ln (\frac { 48 }{ \gamma})$.
> > > Now setting $t = O(\exp( \exp( \frac{nd}{\gamma^2})))$, it suffices
> > > to set $m = O( \max{ \frac{d^2 {(\exp(\frac{nd}{\gamma^2}) + \ln(n))}^2}{\gamma^8} , \ln^2(\frac{n}{\delta \gamma}) \gamma^{2 - 2d} })$ to satisfy both $\\| \bar{p} \\|\_{\infty} $ constraint (line 260) and the width requirement to invoke lemma B.1. This instantiation of the parameter choices satisfies the requirements of theorem 3.1 to obtain the $F_1$ margin (second margin guarantee of theorem 3.1).
> > >
> > >
> > > 2. We apologize for the lack of rigor and will update the paper to include a rigorous discussion about $C_1$ and $C_2$. As discussed above, in the context of 2 layer ReLU networks, $C_1 = 1$ and $C_2 = 0$ ( line 1252 - 1261 in the appendix provide a rigorous proof of this fact ). There is a typo in the current version of the paper where we comment that $C_1$ can be made arbitrarily small for shallow networks (line 275) which should be deleted. Furthermore, the discussion regarding NTK was indeed highly non-rigorous and was mainly based on intuition. We will provide rigorous proofs that $C_2$ goes to 0 for more complicated architectures, namely for normalized deep feed-forward networks and transformer models.
> > >
> > >
> > > 3. Fortunately, it is not the case that the constraint on $\\| \bar{p} \\|\_{\infty} $ is an implicit assumption on the data. As discussed in (1), we *construct* a good reference solution $\bar{p}$ such that  $\\| \bar{p} \\|\_{\infty}  \leq \frac 1 {\sqrt{m}}$. Hence, by taking the width $m$ large enough we can satisfy the norm constraint (line 260). We refer the reviewer to (1) and lemma B.1 for a more in-depth discussion.

---

### Official Review · Reviewer_pv2i · 2024-11-05

**Soundness:** 4
**Presentation:** 3
**Contribution:** 3
**Rating:** 6
**Confidence:** 3

**Summary:**

This paper shows how any architecture satisfying 2-homogeineity and a few regularity conditions on the gradients of the inner layers obtain large margins and low test error. Instantiating this framework to shallow normalized ReLU networks, the paper gives an end-to-end result showing that it obtains large margin and low text error. As a corollary, the paper obtains good test error for k-bit parity problems.

**Strengths:**

1. The instantiation of the framework to shallow normalized ReLU network is novel and interesting. This improved upon results in [Chizat and Bach 2020] to be quantitative, though the computational complexity may be double exponential.

2. For result has concrete implications to the support-only k-parity problem, demonstrating neural networks could learn support-only k-parity much more sample-efficient than kernel methods.

**Weaknesses:**

The first set of results (Theorem 1.1) are a bit abstract and contain non-varifiable assumptions. It is not clear whether it is useful beyond the more concrete shallow normalized ReLU network.

**Questions:**

See weakness

---

> ### Author Response · Authors · 2024-11-27
> **Rebuttal by Authors**
>
> We thank the reviewer for their time and comments.
> We address the reviewer’s question below.
>
> We would like to emphasize part 2 of Theorem 1.1
> (specifically the claim regarding nondecreasing smooth margins) hold for architectures beyond
> shallow normalized ReLU networks such as ResNet and transformer models.
> For ResNet, it is also true that the loss will converge to 0 and the norm of
> the homogeneous parameters, namely the final linear layer and the affine layer from
> the last Batch Norm layer, converge to infinity as the hidden output of ResNet right before the 2-homogeneous head is bounded thanks to Batch Norm.

---

### Official Review · Reviewer_NazB · 2024-11-09

**Soundness:** 3
**Presentation:** 3
**Contribution:** 3
**Rating:** 6
**Confidence:** 3

**Summary:**

This paper proposes an expansion of theoretical results about homogeneous networks to those that only *end* in a 2-homogeneous network (with inhomogeneous layers before that). This includes the majority of common present-day architectures, which end in the combination of a normalization layer with learnable scales and shifts by another linear layer. Under certain assumptions, the paper uses a correspondence of gradient descent on these parameters to mirror descent in a re-parameterized space, showing that gradient flow yields large margins, for a certain definition of margin mostly aligning with prior work.

**Strengths:**

The problem addressed by the paper is important. Identifying that the combination of a normalization layer with a linear layer at the end of these networks might be important to their optimization seems like it might be an important observation. The framing of the particular approach with the connection to mirror descent is clever.

Unfortunately I did not have time to carefully follow the proofs in the supplementary material, but the proof sketches in the main body are clear and the approaches seem reasonable to me.

**Weaknesses:**

I found some of the initial discussion a little misleading in that e.g. the discussion around line 140 made me think that the analysis would apply to arbitrary homogeneous subsets of parameters $\hat w$. While this is true for the margin definition given there, almost all of the paper applies only to the doubled-linear-layer structure of equation (2). This is fine; I just found that bit of discussion confusing, particularly since taking e.g. only the last layer would also yield a 1-homogeneous structure, and I found myself wondering if it would really be true that it would increase the margin defined in terms of each possible homogeneous part of the network. (To be clear, I don't think any statements in the paper are wrong, I just found this brief discussion confusing.)

Much more importantly: if the feature map $F$ were frozen, the setting of this paper would become a "diagonal network" of depth two, for which the implicit bias of gradient descent was already determined by [Gunasekar et al. (NeurIPS 2018)](https://arxiv.org/abs/1806.00468) to be, indeed, an l1 minimizer. The difference with the present paper's results is that $F$ is not required to stay frozen, and of course a frozen $F$ would be a substantial difference from actual training. The setting of the present paper, though, requires that although $F$ can change, it must satisfy "slow-growing derivatives" and "initial good features." Together, this seems to allow the optimization on the last two layers to basically pick out the good features before they change too much to no longer be good. This is interesting, but perhaps not as interesting as it might seem without citing the (here-uncited) previous work for frozen $F$, and with the caveats about initial good features only coming several pages into the paper.

The paper also implicitly argues that the 2-homogeneous structure at the end of transformer-like architectures is very important to their optimization geometry. To be more convincing on this point, since the theoretical results are admittedly a "sanity check," it would be interesting to also run some experiments at least on toy problems to see if this seems to line up. That is, do people leave the learnable parameters on in their normalization layers that are immediately next to a linear layer because it actually helps, or just because they didn't think about it?

**Questions:**

- In the case of frozen $F$, would direct application of the results of Lyu and Li plus Gunasekar et al. give roughly the same results as you obtain?

- Is my characterization of "slow-growing derivatives" + "initial good features" -> "the features can't break before last-layer optimization finds the good ones" correct?

- I'm curious about the relationship of your "initial good features" assumption to the "initial alignment" results from the series of papers by Emmanuel Abbe and coauthors, particularly [Abbe et al. (ICML 2022)](https://arxiv.org/abs/2202.12846). It seems, without thinking about it very much, that your "initial good features" assumption implies a high "initial alignment"; as they show, this is in some sense necessary for learning, but in a stronger form than they make.

---

> ### Author Response · Authors · 2024-11-27
> **Rebuttal by Authors**
>
> Thank you for the valuable comments and insightful questions. We address your
> questions below.
>
> 1. Unfortunately, it does not seem possible for the reasons stated below. Firstly, it should be noted that both Gunasekar et al. and Lyu&Li are truly asymptotic results and it is not clear how to get finite time guarantees using a combination of Gunasekar et al. and Lyu&Li.
> In addition, both Gunasekar et al. and Lyu&Li operate in the late regime where perfect classification has been achieved and the model has small positive margin. In fact the first half of the proof of theorem 3.1 is dedicated to proving that one achieves a small positive margin. Furthermore, the convergence to the max margin predictor in
> Gunasekar et al. further assumes both the iterates produced by gradient descent and the loss gradients converge in direction, a fact that is highly non-trivial to prove.
>
> 2. Yes, that characterization is correct. However, we would like to emphasize that for shallow ReLU networks, we prove both the “slow-growing derivatives” property and the existence of “initial good features” under the distributional margin assumption in section 4.
>
> 3. While related, the strength of the assumptions are somewhat incomparable. Indeed, in the setting where perfect classification is impossible (e.g. data distributions where inputs have nonzero probability of being labeled both positive and
> negative), one can still have “high alignment” but the “initial good features” assumption is violated as the margin cannot be positive, hence not separable. However, on the flip side, the initial “good features” assumption can still hold but the “initial alignment” can be very small. This does not contradict the hardness results in Abbe et al. as getting a large margin (which is roughly equivalent to learning the target function) can take exponential time (in fact in the worst case it can be doubly exponential).
>
> >I found some of the initial discussion a little
> >misleading in that e.g. the discussion around line 140 >
> >made me think that the analysis would apply to arbitrary
> > homogeneous subsets of parameters
> >. While this is true for the margin definition given
> >there, almost all of the paper applies only to the
> >doubled-linear-layer structure of equation (2). This is fine
>
> We apologize for the confusing discussion. The margin definition was
> described in full generality as the second part of theorem 1.1 handles
> arbitrary homogeneous subsets of parameters. We have revised the discussion
> following theorem 1.1, clarifying that part 1 of theorem 1.1 only applies
> to architectures with a double-linear-layer head.
>
> >I just found that bit of discussion confusing,
> >particularly since taking e.g. only the last layer would
> > also yield a 1-homogeneous structure, and I found
> > myself wondering if it would really be true that it
> > would increase the margin defined in terms of each
> > possible homogeneous part of the network. (To be clear,
> > I don't think any statements in the paper are wrong, I
> > just found this brief discussion confusing.)
>
> Surprisingly, the *smoothed* margin defined in terms of any homogeneous subset $ \hat w$
> is nondecreasing as asserted in theorem 3.6. In fact, this claim does not need any bounded assumptions of $ h $ and its gradients with respect to the ignored parameters. This is crucial as
> any homogeneous parameters (of nonzero degree) in the ignored subset $ w \ \hat{ w }$
> violate the boundedness assumptions. However, as discussed in the section containing the margin definition,
> the notion of margin (and smoothed margin) only makes sense when the margin does not grow arbitrarily
> with the norm of the ignored parameters $ w \ \hat w$.
>
> If the reviewer is not satisfied with this response, can they please clarify their further concerns?

---

> > ### Comment · Reviewer_NazB · 2024-11-30
> >
> > Thank you for your response, which has addressed my concerns (and the other reviews didn’t raise any substantial new ones for me). I still think this paper doesn’t give as complete of an argument as it ideally could (mainly, I would have really appreciated some empirical experimentation on toy problems exploring the importance of this effect), and so will keep my numerical score.

---

### Author Response · Authors · 2024-11-27
**Revision Summary**

We thank the reviewers for their comments and time. In order to improve clarity,
we have revised the paper and the main changes are as follows.

* We provided a more thorough explanation of the uniqueness of 2 homogeneity
following the statement of lemma 3.2. As a summary, only a 2 homogeneous head induces
a reparameterized flow that explicitly optimizes the superior $F_1$ space as opposed
to the $F_2$ space.

* In response to concerns regarding the novelty of theorem 3.6,
we revised the discussion following theorem 3.6 by providing a more in depth comparison with https://arxiv.org/pdf/1906.05890 with regards to generality of the result as
well as proof techniques.

---

### Meta-Review · Area_Chair_AjRt · 2024-12-22

**Metareview:**

This paper considers the implicit bias of neural networks where the last two layers form a two-layer linear diagonal network; such networks can be motivated from networks having normalization layers with post-normalization diagonal scaling parameters in their last hidden layer. Under technical assumptions that can be roughly described as "slow-growing derivatives" and "initial good features,” the paper proves that the network achieves a good normalized margin early in training, and then the smoothed margin is non-decreasing until the end of training. The first results are given for an abstract family of networks (Section 3), while Section 4 presents a more concrete instantiation of the theorems under a 2-layer normalized ReLU network and an additional assumption on the data distribution.

The paper brings up interesting viewpoints, focusing on the common structure of the last two layers in networks employing normalization layers. The main results are developed for a general framework and they can be specialized to more concrete results for two-layer normalized ReLU networks and for sparse parity problems. Many reviewers, and myself, found the proof technique novel and interesting. The technique to view the evolution of $a \odot b$ as mirror flow on $u$ and using online convex optimization to handle the time-dependency of $V(s)$ is intriguing.

However, the main criticism from all reviewers came from the fact that the main result (Theorem 3.1) relies on complicated and hard-to-verify assumptions and conditions. The “initial good features” assumption requires that the initialized parameter $V(0)$ already yields linearly separable hidden representations $F(x_i;V(0))$. The “constants” $C_1$ and $C_2$ are in fact dependent on the entire trajectory up to time $t$, which makes it impossible to verify a priori. With $t$ growing double exponentially, the conditions on $\\lVert \\bar{p} \\rVert_\infty$ and $C_2$ also become restrictive. Although the authors demonstrate an example in which the theorem can be applied, the application requires unrealistically large width and training time (albeit not infinite). These strong and hard-to-verify conditions make me worry if the results will really be useful/applicable in other setups.

Also, it looks to me that the presentation has some room for improvement, including:
- As noted, simplifying or removing the assumptions and conditions in the main theorem will improve the paper.
- More concrete examples beyond 2-layer normalized ReLU networks will be helpful.
- As Reviewer NazB suggested, the authors could consider running some (toy) experiments corroborating the theory.
- The paper looks difficult to follow if the reader doesn’t have prior knowledge on $F_1$ and $F_2$. They are not formally defined in the paper, except at the very end (in Line 500). Some formal description of these classes should be given.
- The paper claims that the normalized ReLU networks can be viewed as an idealization of BatchNorm, but I question why this network bears resemblance to BatchNorm. There is no batch-wise normalization and the input is not centered at zero.
- The authors claim regarding Theorem 3.6 that any model $h$ that has a subset of homogeneous parameters satisfies the theorem, but one needs to also mention the condition $\sup_\theta \sup_{\\lVert \hat w \\rVert \leq 1} \\lVert \nabla_{\hat w} h(x; (\hat w,\theta)) \\rVert < \infty$ because the claim can be misleading when stated without the condition.
- As noted by Reviewer NazB, the implicit bias of linear diagonal networks on binary classification was studied in Gunasekar et al (2018). Of course, the paper goes beyond the setting of this work, but the authors must contextualize the current submission with respect the this existing result and its subsequent follow-ups.

Overall, although the review scores were borderline positive, I believe the paper sits slightly below the acceptance threshold and requires some major revision before getting published. At this time, I recommend rejection.

**Additional Comments On Reviewer Discussion:**

N/A. I already put all additional comments in the main meta-review.

---

### Decision · Program_Chairs · 2025-01-22

Reject